


# A novel data-driven analytical framework on hierarchical water allocation integrated with blue and virtual water transfers

Liming Yao[1,2], Zhongwen Xu[1], Huijuan Wu[3], and Xudong Chen[1]

[1]Business School, Sichuan University, Chengdu 610064, China
[2]State Key Laboratory of Hydraulics and Mountain River Engineering, Sichuan University, Chengdu, 610064, China
[3]Institute of Water Policy, Lee Kuan Yew School of Public Policy, National University of Singapore, 469A Bukit Timah Road, 259770, Singapore

**Correspondence:** Xudong Chen (chenxudong198401@163.com)

**Abstract.** This study proposes a novel data-driven analytical framework to determine optimal allocation and conjunctive strategies of blue and virtual water transfers in the presence of different hydrological and economic conditions. Due to the water scarcity and uneven distribution, a Stackelberg-Nash-Harsanyi equilibrium model is developed to cope with the hierarchical conflict between the water affairs bureau and multiple water usage sectors. Results show that coalitional strategy of blue

and virtual water transfers can substantially save water and improve utilization efficiency without harming sectors' benefits and increasing ecological stresses. Under various polices, we use data-driven analysis to simulate hydrological and economic parameters, such as available water, crop import price, water market price etc. Different results obtained through adjusting hydrological and economic parameters show that the optimal allocation and transfer strategy is more sensitive to hydrological factor instead of economic factor. Moreover, the results prove that conjunctive blue/virtual water transfers can response to

market fluctuation. Ultimately, the proposed framework can achieve a sustainable management for physical and virtual water supply system under future hydrological and economic uncertainty.

## 1 Introduction

The uneven spatial distribution of water, rapid population growth and rapid economic development have exacerbated water scarcity and stimulated regional and sectoral conflicts around the world. Moreover, extreme climatic and hydrological conditions have increased the pressure caused by water scarcity. Several provinces, mostly located in northern China, suffer from severe water scarcity almost 7 months a year (Zhuo et al., 2016; Ma et al., 2006; Cai, 2008). Furthermore, a relatively high level of crop provisioning in southern China aggravates the problem in northern China (e.g., Xinjiang, Heilongjiang, Guangxi,

Hunan, Hebei, and Inner Mongolia) (Wang et al., 2014). This pattern has led to a paradox in which water-intensive crops are exported from water-insufficient northern China to water-rich southern China, while water resources must be transferred



from water-rich southern China to water-scarce northern China to enable crop production. The South-to-North Water Transfer Project was used to reduce northern China water shortages; however, from a long-term perspective, the project is extremely costly and easily leads to the destruction of the ecological environment (Chen et al., 2017). In light of the above problem, we

study two important issues focused on irrigation district problem, namely, blue water transfer as a means of reallocating water among sectors to decrease water usage vulnerability, and virtual water transfer to decrease the water stress and modify the water usage structure under the international trading environment.

The district irrigation in many countries and regions, usually consists of two different hierarchical structures of a water affairs bureau and water usage sectors within an irrigation district. Due to multiple objectives to multiple hierarchical decision

makers (water users and water affairs bureaus), inevitable contradiction appears. To solve water conflicts, stochastic dynamic programming (Zeng et al., 2017; Wong et al., 2012), multi-objective programming, and game theory (Madani, 2010; Yu and Hong, 2017; Ye et al., 2018) used to optimize the problem. To be specific, multi-objective programming models have been used to solve conflicting objectives in water allocation problems (Xu et al., 2018); however, these models have neglected dynamic feedback mechanisms possibly existing in different hierarchical structures, resulting in solutions that do not maximize benefits

at the system scale. Instead, game theory, which originated with the pioneering work of (Neuman and Morgenstern, 1944), was a successful alternative tool used for analyzing strategic interactions among different hierarchical decision makers. Based on a general overview, we found the successful application of game theory in different fields. We recall some typical game-theoretical models from water resource management relating to our work, as shown in Table 1. In a water allocation context, consider the conflict of two hierarchical stakeholders through strategic interaction is vital. A Stackelberg game is widely used

to cope with a kind of game with stakeholder in different position, besides this approach provides a "soft path" for balancing the individual- and system-level benefits. Hence, the idea Stackelberg game is proper to solve the water allocation problem.

Because there are multiple followers, one of the major problems is how to allocate limited water resources to various water usage sectors in a sustainable way, including agricultural, industrial, domestic and ecological sectors, another is whether it can transfer the excess water to other sectors. Under government guidance, sector usage managers within a region have the right

to reallocate surplus water, and other sector managers can purchase these additional water resources, with the transfer price decided from negotiations between the transfer participants. Multi-objective programming has been extensively used to resolve water allocation conflicts of competing water usage sectors (Sedghamiz et al., 2018a; Babel et al., 2005; Brown et al., 2015; Madani, 2011). However, it cannot reflect the strategic interaction among competing stakeholders at lower levels. Therefore, Nash (1953) suggested a non-linear optimization model to determine an optimal solution to the 2-player bargaining game

over sharing a resource under cooperation and to enforce a fair and efficient allocation of the resource among the rational bargainers. Later, Harsanyi (1959, 1963) generalized the Nash solution for a 2-player bargaining game to an $n$-players game, which was more suitable for the multiple stakeholder non-cooperative situation. The Nash-Harsanyi equilibrium was found to be beneficial to all the stakeholders involved by strategic interaction, according to Sedghamiz et al. (2018b). Overall, during the water transfer process, an $n$-person Nash-Harsanyi bargaining model is developed, which also considers the water withdrawal

and reallocation process among sectors. To the best of our knowledge, the applications of game theory to solve two different types of conflicts within an irrigation district that considers the leader (water affairs bureau) in the dominant position and the



followers (water usage sectors) are comparatively few. Hence, there is an urgent need to propose a suitable bilevel model to solve the practical problem.

Recently, water market has been one of the key approaches for improving water utilization efficiency (Wang, 2018). For an
irrigation district in northern China, the implementation of agricultural water conservation and transfer projects ensured that the agricultural sector had the right to make decisions on water allocation to crop irrigation or blue water transfers (known as water market approach, which involves the right to short-term use of water) (Dai et al., 2017; Ahmadi et al., 2019). Water transfers gradually achieve a win-win situation, where enterprises obtain water consumption rights, and the irrigation area solves the problem of insufficient funds when carrying out agricultural water conservation projects. In this regard, all endeavors and
activities pertaining to water scarcity problems in China have revealed a strong requirement to formulate a comprehensive and sustainable national water resources plan that must be endorsed in the twenty first century. However, only considering blue water transfer among sectors solve the water scarcity problem fundamentally, instead alleviate the economic loss caused by water scarcity.

If blue water transfer realise water transfer among sectors, then virtual water transfer (content in crops, such as soybeans,
rapeseed, cotton, and barley) (Zhuo et al., 2016) solves water transfer among countries. It is possible that importing water-intensive goods (particularly crops), rather than producing them domestically, could conserve water resources and drive economic development in exporting countries (Shtull-Trauring and Bernstein, 2018; Jiang and Marggraf, 2015). Virtual water, first introduced by Allan et al. (1993), is the water embedded in a product in a virtual form Liu et al. (2009). The virtual water transfer, therefore, allows for the redistribution of water resources between countries, which means that water-scarce countries
can conserve their own water resources, and water-sufficient countries can obtain greater economic benefits by transferring the surplus water (Liu et al., 2015a, b) to water-scarce countries. In this study, we only calculate the virtual water contents of import crops. Besides, we consider blue and virtual water together within the irrigation district to completely solve the water scarcity problem.

Blue water transfer has the ability to reduce economic loss caused by unsuitable water supply, and virtual water could be used
as a viable substitute for improving water utilization efficiency and alleviating water scarcity. However, it is worth noting that an increase in crop imports would lead to a decrease in local production, which could significantly reduce agricultural incomes for the locally produced crops. Similarly, an increase in crop exports could lead to local water and ecological stresses. To the best of the authors' knowledge, despite some progress obtained in previous studies (Lamastra et al., 2017; Mohammadikanigolzar et al., 2014; Liu et al., 2015a; Duarte et al., 2016; Zhuo et al., 2016; Chen et al., 2017), there is still a lack of a systematic
method for determining the optimal amount of virtual water transfers. One of the main challenges is finding how much water moves in and out of the region and then determining how to best benefit from this water. However, the optimization of virtual water transfer research is particularly focused on crops that require a great deal of water, such as cotton or rice. In addition, few articles have considered blue/virtual water transfers together in an optimization model, which led to the current ineffectiveness of water rights allocations. Hence, the first objective is to determine the optimal water allocation strategy by incorporate the
idea of blue/virtual water transfers to provide a new, unstructured alternative to satisfy the sectoral water demands. Overall, in addition to accurately quantifying blue water transfer among sectors, this paper determines the optimum quantity of imported



crops required to conserve water in the importing country and the optimum quantity of exported crops required to ensure water sufficiency in the exporting country.

A second innovation is study the game theory, which has very little literature on irrigation districts taking blue/virtual water
transfers and water allocation into consideration within a bilevel framework at the same time, and the main points on this topic are listed in Table 1. Due to this problem's unsuitability for modeling by conventional methods, a novel game model is presented to consider water allocation and blue/virtual transfers together, in view of the hierarchical structure of the problem. Both hierarchical decision makers have different decision preferences. Sedghamiz et al. (2018a) considered the objectives that consisted of equity maximization, agricultural benefit maximization for each region, maximization of green water utilization
and minimization of environmental shortages. Qi et al. (2018) maximized the water requirement satisfaction index of agents at the superior level and constructed a subordinate model based on the Nash bargaining model. Wang et al. (2008) aimed to model equitable and efficient water allocation among competing users. However, most of these models aimed to maximize the economic returns to various water users and did not consider the satisfaction ratio of the amount of water supplied to the normal demand of a particular water user for a sustainable water allocation system. However, few optimization models
have been developed. To fill this gap, this paper fully considers the sustainable development in each sector and minimizes the objective function of water vulnerability, which consists of destroying the degree of economic loss, with the purpose of meeting the "water demands" and giving effective water withdrawal guidance to the water usage sectors. In addition, to ensure both economic benefits and water demand satisfaction in different sectors, the water affairs bureau seeks to maximize the water resource system water utilization efficiency, with the incorporation of blue/virtual water transfers. Based on the complex
conceptual framework, a novel Stackelberg-Nash-Harsanyi equilibrium model is developed to optimize realistic water resource management. In this way, an equilibrium can be gained by strategic interaction among the water affairs bureau and water usage sectors; moreover, competing water usage sectors not only make decisions targeting their own benefits from the sustainable perspective but also allow blue water transfers to improve the system's water usage efficiency. As we know, bilevel optimization techniques are intrinsically complex to solve (Qi et al., 2018; Hossein et al., 2018; Wei et al., 2017). In regard to a solution,
Eichfelder (2010) presented several new theoretical results for general multi-objective bilevel optimization problems using the optimistic approach. In other studies, different methods, such as particle swarm optimization and artificial neural networks, have been used to solve bilevel decision-making problems in water exchange in eco-industrial parks (Ramos, 2016), product engineering (Liu et al., 2017b), and lot-sizing problems (Wong et al., 2012). However, these techniques have seldom been applied to practical cases for the allocation of water resources due to their complexity. In this paper, the bargaining weights
method, Nash-Harsanyi solution method, and genetic algorithm (GA) are combined to solve the proposed model.

In China, an irrigation district is a typical unit within which agricultural, domestic, industrial and ecological sectors compete for water resources. We choose the Hetao irrigation district as a case study because it is a severe water-scarce district that provides grain supplements to southern China. Based on real-world practice, a bilevel framework for water resource allocation consisting of two types of decision makers, i.e., the water affairs bureau and three water usage sectors, is developed. In
addition, this paper considers water market inter-regional and international transactions (down to virtual water transfer) and then incorporates and quantifies water transfers in the form of blue water or virtual water when optimizing the water allocation





**Table 1.** Literature review

| Articles | Problem statement | Methodologies | Difference in technical strategy point |
|---|---|---|---|
| Dai et al. (2017) | The compensation mechanism for agricultural water transfer | Classification theory | |
| Jiang and Marggraf (2015) | Assessing the virtual water transfer in agricultural products between Germany and China | Statistic analysis | Ignoring quantitative analysis |
| Shtull-Trauring and Bernstein (2018) | Analyzing virtual water transfer on a country level | Statistic analysis | |
| Ahmadi et al. (2019) | Optimizing Beheshtabad Water Transfer Project | Both cooperative and non-cooperative approaches | |
| Fu et al. (2018) | Water allocation | Two-stage model considering Nash-Harsney equilibrium model | Ignoring virtual/blue water transfers and hierarchical strategic interaction |
| Xu et al. (2018) | Water allocation | Multi-objective programming model | |
| Sedghamiz et al. (2018b) | Water and crop area allocation | Leader-follower game | |
| Sedghamiz et al. (2018c) | Water management with the presence of executive managers in top-level and the agricultural sectors in low-level as leader and followers | Two-level optimization model | Ignoring virtual/blue water transfers and conflict at the lower level |
| Guo et al. (2012) | Solving the multi-reservoir operation problem in inter-basin water transfer supply project | A bilevel model | |
| Sedghamiz et al. (2018a); Qi et al. (2018); Wang et al. (2008) | The objectives consist of equity maximization, benefit maximization | Multi-objective programming model | Ignoring virtual/blue water transfers, strategic interaction and the objective of sustainable development |

process. To accomplish this objective, the Stackelberg-Nash-Harsanyi equilibrium is obtained. Overall, the main results and contributions are summarized as follows:

*Contribution 1*: Having incorporated the concepts of blue and virtual water transfers, our model is able to further relieve
the water scarcity stress and offers insights on crop planting and import/export quantities. We find different strategies faced with changing hydrological and market conditions. In the spirit of the water stress index, water usage efficiency and sectoral vulnerabilities, we ascertain that international transaction saved water and land. The water usage efficiency increased because of crops planting plan's modification and water market among the three sectors. Besides, blue water transfer provided an opportunity for each sector to achieve an efficient utilization of water.

*Contribution 2*: A novel game-theory model based on the Stackelberg game and Nash-Harsanyi equilibrium is developed for resolving the "leader-followers" and "competing followers" conflicts by strategic interaction. We find that the total consumption solved using the two-stage optimization model is mostly higher than that for the proposed model, which is detrimental to save water.


## 2 Study area and water trading background

### 2.1 Study area

Hetao irrigation district, which is an area that has recently been confronted with a contradiction between North-South crop export requirements and water scarcity. Irrigation districts in China produce more than 75% of the grain consumed in China and have become increasingly important in ensuring the safety of both China's food supply and its socioeconomic development (Wang et al., 2005). As studies on small and specific irrigation districts have been more significant than national studies in terms of resolving water scarcity problem, the Hetao irrigation district is chosen as the case study to resolve the following problems: (1) optimize the water withdrawals in agricultural, domestic and industrial sectors and determine the irrigation requirements for different crops in a planning year; (2) optimize the virtual water quantities to be imported and exported under the international trading environment; and (3) optimize the blue water quantities to be transferred under the water market environment. This study provides perspectives on water reallocations and can assist in alleviating local water scarcity and promoting sustainable development. Hetao irrigation district is an agricultural production and trade area in China with an irrigated area of $5.74 \times 10^3$

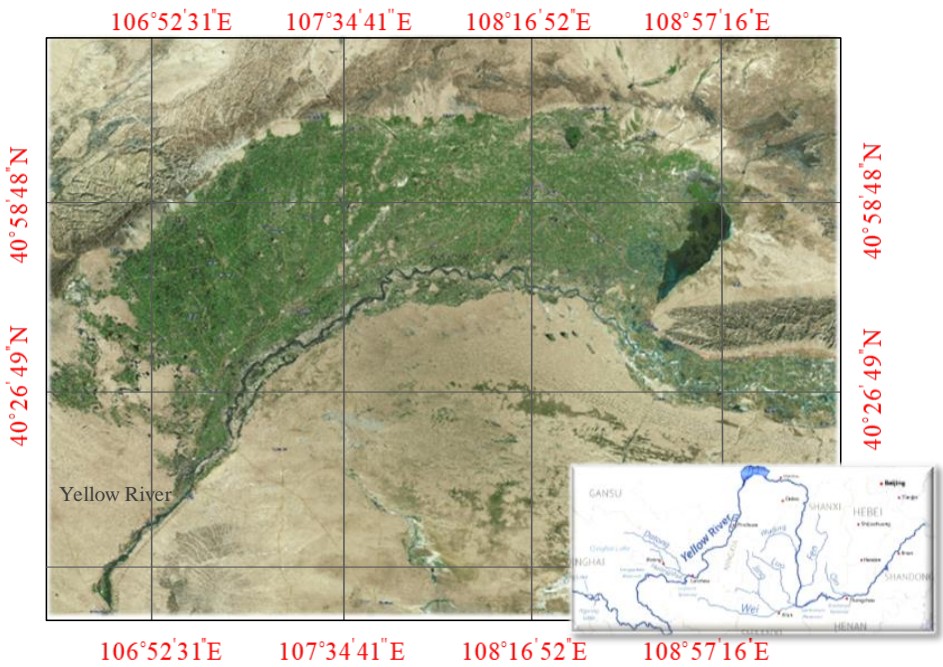

**Figure 1.** Hetao irrigation district longitude-latitude projection image ©Google Maps

km$^2$, with the irrigation water being mainly supplied by the Yellow River. The district is located in western Inner Mongolia, China (40°13'-42°28'N, 105°12'-109°53'E) and includes five counties (Dengkou, Hanghou, Linhe, Wuyuan, and Qianqi) (Fig. 1). The general problems confronting the Hetao irrigation district are the increasing water requirements and the severely constrained freshwater resources. Agricultural activities in this region consume approximately 93 % of total regional water





consumption Feng et al. (2012). The district has a continental monsoon climate. Rainfall is scarce (130-215 mm y$^{-1}$) and erratically distributed (70% in July, August, and September), and the annual evaporation is 2100-2300 mm Liu et al. (2015a, b); Wang et al. (2005). Being close to port cities, the manager of the Hetao irrigation district has the privilege of importing key crops from Russia and other countries, which helps relieve the water consumption stress.

## 2.2 Water trading background

Under water trading background, both blue and virtual water trading are considered. Blue water is the surface or groundwater that evaporates during a production process, and virtual water is the volume of water necessary to produce a certain commodity (Allan et al., 1993). Blue water can be directly transferred from one sector to another through conveyance infrastructure after each sector (e.g., domestic, industrial or agricultural) has been granted temporary water withdrawal rights, and virtual water can be indirectly transferred from one country to another. The virtual water related to the crops traded is calculated based on

the sum of the volumes of crop exports or imports. Three types of crops are included in this study: wheat, maize and sunflower.

### 2.2.1 Transaction Prices

Water pricing is a key component of current water policy reforms in China, with the belief that reasonable pricing reform can improve water transfer, in other words, transaction prices can control water usage and market participation (Erfani et al., 2014).

In previous studies, most scholars choose to define the water trading price based on supply and demand data. On the one

hand, there is insufficient data to fit out the determined relationship, on the other hand, this kind of pricing mechanism based on supply and demand in the complete market cannot guarantee the effective allocation of scarce water resources in the long term. In fact, water rights trading usually only takes place between two participators, which should depends on the negotiated price that both participators are willing to accept. Therefore, in this study, the unit price of water trade should be between the buyer's willingness to pay and the seller's reservation price (Erfani et al., 2014). Thus the price is determined as the weighted

sum of willing price to pay and reservation price to sell, which is called a negotiated price.

$$\text{PTI} = \theta p_1 + (1 - \theta)p_2, \quad \text{PTD} = \theta p_1 + (1 - \theta)p_3, \tag{1}$$

where $0 \leq \theta \leq 1$, $p_i$ is the water price set by the leader for water withdrawal, PTI is the price of water transferred to the industrial sector, and PTD is the price of water transferred to the domestic sector.

## 3 Methods

### 3.1 Bilevel water allocation system framework

Fig. 2 gives a complete description of the fours steps covered in this paper. First, the stakeholders in the water allocation system are identified, after which a conceptual water allocation framework is constructed (Fig. 3). The proposed model is then applied to a real-world case study to make decisions on water allocations, crop irrigation, and the international import, inter-regional



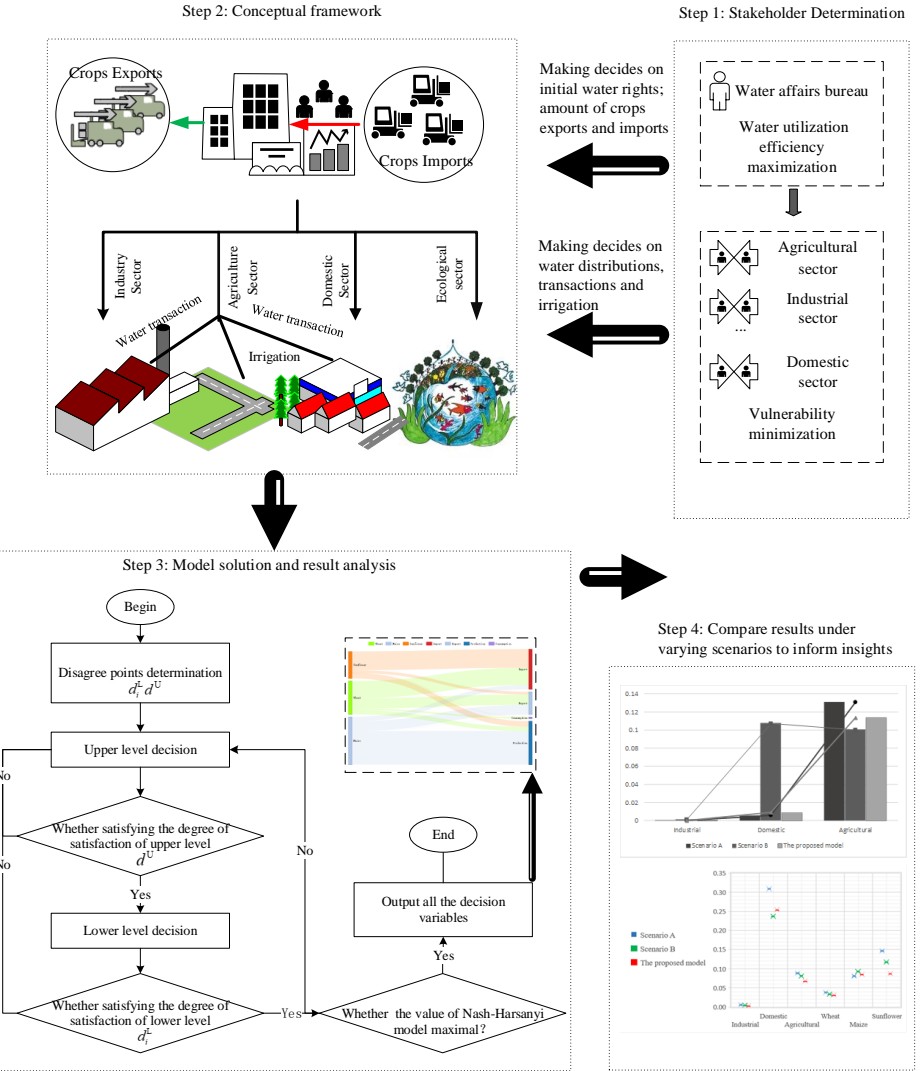

**Figure 2.** Four steps in the water allocation plan and future solutions

export and blue water transfer quantities. Finally, to identify an appropriate approach for future water sustainability, multiple

scenarios are examined.

Driven by a traditional water resource management process (Harsanyi, 1963), this paper constructs a bilevel framework with one leader and multiple followers, as shown in Fig. 3, in which the water affairs bureau is the leader and the water usage sectors are the followers. Two games are included in the framework: a Stackelberg game between the leader and followers in which the water affairs bureau (the leader) has the leading role in allocating the water resources, moves first and has complete

information about the followers' possible reactions, after which the water usage sectors (the followers) react after being given information about the leader's announced strategy; and a Nash-Harsanyi game between multiple followers (Fu et al., 2018),





which is an evolved cooperative game, for which an $n$-person Nash-Harsanyi bargaining solution is studied after which the $n$ followers make corresponding decisions based on the decisions of the leaders.

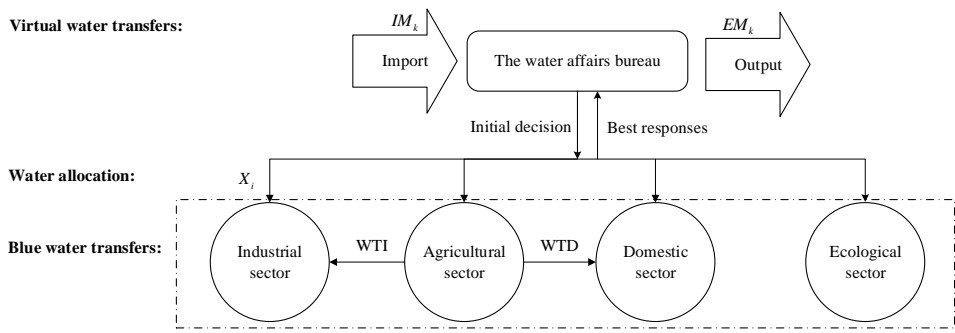

**Figure 3.** Bilevel water allocation system framework

## 3.2  Global Model

This section outlines the mathematical model formulation. Water resource management allocation in irrigation districts usually involves a single leader and several followers, which is suitable for constructing a Stackelberg-Nash-Harsanyi equilibrium model. Then, the model is developed based on the following assumptions:

*Assumption 1:* Regions with water deficits can import agricultural products from neighboring countries that have surplus water.

*Assumption 2:* The water market is considered for all sectors, with the trading price being defined as the average willingness to pay of the partners.

*Assumption 3:* In semiarid and arid regions water is scarce relative to land. So we assume land availability does not constrain crop decisions.

Because of uneven distribution of water and changing environment, the key problem is to reallocate the limited water resources to competing water usage sectors while maintaining a system balance that maximizes the overall efficiency and minimizes each sector's vulnerability. In contrast to the traditional two-stage optimization model, stochastic dynamic programming Zeng et al. (2017); Wong et al. (2012), and multi-objective programming Madani (2011), this study considers a strategic interaction from the hierarchical perspective, which helps to solve different kinds of conflicts existing in water resources management systems. At present, there is very little literature on irrigation districts taking blue/virtual water transfers and water allocation into consideration within a bilevel framework at the same time. Hence, a novel Stackelberg-Nash-Harsanyi equilibrium model (2) is developed by integrating (D4)-(D21) in Appendix to consider water allocation and blue/ virtual transfers together, in view of the hierarchical structure of the problem.





The objective functions at the upper and lower levels are to maximize the water utilization efficiency and minimize the
sectoral vulnerability. At the upper level, the first constraint promises the water withdrawal for the three sectors cannot exceed

the initial water gained by the irrigation district; the second constraint stipulates that transaction price should not exceed the
water withdrawal price; the third constraint guarantee the minimum ecological water requirements; the forth constraint require
that the annual export volume plus the grain consumption should be smaller than the total grain yield plus the annual import
volume. At the lower level, the constraints for each follower is that the value of water withdrawal is non-negative. For the
agricultural sector, an additional constraint is that the total planting areas don't exceed the total area of this irrigation district.

It is worth noting that in semiarid and arid regions water is scarce relative to land. So we assume land availability does not
constrain crop decisions (Zhu et al., 2015).

$$
\max \ \mathrm{Eff} = \frac{\mathrm{Re}}{\mathrm{Cons}}
$$

$$
s.t. \begin{cases}
\sum_{i=1}^{4} X_i \leq \mathrm{AW} \\
p_1 < \mathrm{PTI} < p_2 \\
p_1 < \mathrm{PTD} < p_3 \\
X_4 \geq e \\
\mathrm{EM}_k + \mathrm{POP}\varpi_k \leq l_k + \mathrm{IM}_k \\
\min F_{1k} \\
s.t. \begin{cases} \sum_{k=1}^{s} A_k \leq A, A_k = \frac{X_{1k}}{W_k} \\ x_{ik} > 0 \end{cases} \\
\min F_2 \\
s.t. X_2 > 0 \\
\min F_3 \\
s.t. X_3 > 0
\end{cases}
\tag{2}
$$

### 3.3 Solution Procedure

There are two types of games: between the leader and the followers and between the followers. The leader makes decisions
on the water transfers, whether in the form of virtual water or blue water, and the followers decide on their own right to water
withdrawal and irrigation. Therefore, a compromised solution is needed between the upper- and lower-level decision makers.
Fig. 4 illustrates the four steps for the Stackelberg-Nash-Harsanyi bargaining process.

**Step 1:** Determine the disagreement points $(dis^U, dis_i^L)$ and bargaining weights $(\alpha_i)$ for the decision makers at each level;

randomly generate an initial solution.

**Step 2:** Maximize the value $\prod_{i=1,2,3} \left(F_i - dis_i^L\right)^{\alpha_i}$, by selecting the level objective functions that better than their disagreement points; determine if the termination condition is satisfied; if yes, go to *Step 3*, otherwise, continue to add generations.





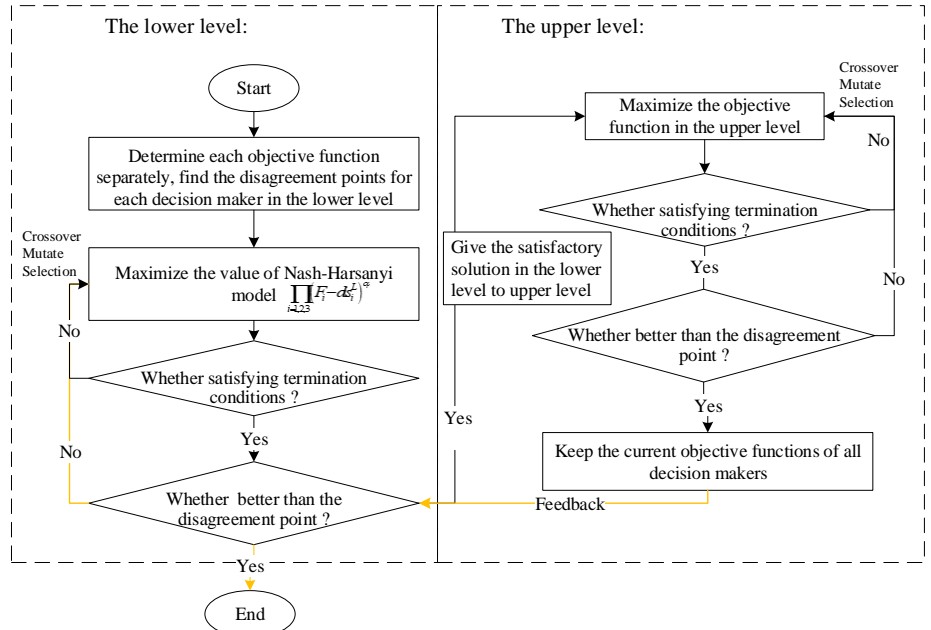

**Figure 4.** Steps for solving the proposed model

*Step 3:* Maximize the upper-level objective function on the premise of it being better than the disagreement point; determine if the termination condition is satisfied. If yes, output all the decision variables, and go to *Step 4*; otherwise, continue to add generations.

*Step 4:* Determine whether the lower level objective function is still better than the disagreement points. If yes, end the loop; otherwise go back to *Step 2*.

Combined with the practical problem described in this paper, we define the vector of the disagreement points as the maximum vulnerability to the followers and the minimum efficiency to the leader. Specifically, we calculate the disagreement point as follows:

The individual best and least solutions of the leader are $(X_i, \mathrm{EM}_k, \mathrm{IM}_k, \mathrm{WTI}, \mathrm{WTD}, x_{1k}; \mathrm{Eff}^{\max})$ and $(X'_i, \mathrm{EM}'_k, \mathrm{IM}'_k, \mathrm{WTI}',$ $\mathrm{WTD}', x'_{1k}; \mathrm{Eff}^{\min})$, respectively, which are obtained by using GA method, where

$$\mathrm{Eff}^{\max} = \max \mathrm{Eff}(\mathrm{X}_i, \mathrm{EM}_k, \mathrm{IM}_k, \mathrm{WTI}, \mathrm{WTD}, x_{1k}) \qquad (3)$$

$$\mathrm{Eff}^{\min} = \min \mathrm{Eff}(\mathrm{X}'_i, \mathrm{EM}'_k, \mathrm{IM}'_k, \mathrm{WTI}', \mathrm{WTD}', x'_{1k}) \qquad (4)$$

Similarly, the best and least solutions of the followers are $(X''_i, \mathrm{EM}''_k, \mathrm{IM}''_k, \mathrm{WTI}'', \mathrm{WTD}'', x_{1k}; F_i^{\min})$ and $(X'''_i, \mathrm{EM}'''_k, \mathrm{IM}'''_k,$ $\mathrm{WTI}''', \mathrm{WTD}''', x'''_{1k}; F_i^{\max})$, respectively, where

$$F_i^{\min} = \max F_i(X''_i, \mathrm{EM}''_k, \mathrm{IM}''_k, \mathrm{WTI}'', \mathrm{WTD}'', x_{1k}) \qquad (5)$$

$$F_i^{\max} = \min F_i(X'''_i, \mathrm{EM}'''_k, \mathrm{IM}'''_k, \mathrm{WTI}''', \mathrm{WTD}''', x'''_{1k}) \qquad (6)$$





Now, the lower tolerance limits $(\text{Eff}^{\min}, \text{F}_i^{\max})$ for achieving the goal levels of the leader and follower can be defined as disagreement points (denoted as $dis^U, dis_i^L$) for the decision makers at each level. Then, the additional constraints for each level were added so that each objective function value was better than the disagreement point respectively ($\text{Eff} > dis^U$ with $F_i < dis_i^L$), namely, the disagreement point presents the worst result that the decision maker was unwilling to accept.

In addition, the bargaining weights, which reflect the degree of importance to each follower, are defined based on water demand elasticity. Considering the demand principle, the formula for calculating the water demand elasticity level is $\delta_i = 1 - \frac{d_i}{\Sigma d_i^L}$, and then the bargaining weight of each follower under demand principle id calculated as $\alpha_i = \frac{\delta_i}{\Sigma \delta_i}$.

## 3.4 Economic parameters

The main data sources for the solution are based on the Bayna Noaoer Yearbook, Hetao irrigation district statistical data and some published papers. The outputs of each sector per unit of water ($\text{ERW}_i$) are $\text{ERW}_1 = 2.34$ RMB/m$^3$, $\text{ERW}_2 = 109.96$ RMB/m$^3$, and $\text{ERW}_3 = 131.17$ RMB/m$^3$ Liu (2016). Referring to the website "http://price.h2o-china.com/" and the Development and Reform Commission and Department of Water Resources' agricultural water price adjustment programs, the price of water is to be $p_1 = 0.103$, $p_2 = 3.85$, and $p_3 = 4.40$ in 2020. The prices of crops (wheat, maize and sunflower) imported from other countries are determined according to the average import prices of agricultural crops over the years, namely, $c_1 = 2.58$, $c_2 = 1.50$, and $c_3 = 4.10$ RMB/kg, respectively. Concerning the water transfer price, $\theta$ is set equal to 0.5; the water transfer prices are PTI $= 1.98$ and PTD $= 2.25$. The transfer cost TC $= 1.00$ RMB/m$^3$ refers to the website "http://cwex.org.cn/lising/". The consumptions of each crop ($\sum\limits_{k=1}^{3} \text{POP}_k \varpi_k$) in 2020 are predicted to be $1.22 \times 10^8$, $1.73 \times 10^7$ and $2.90 \times 10^5$ kg respectively.

## 3.5 Hydrological parameters and water demand analysis

Using the data obtained from the BayanNur Water Resources Bulletin from 2012-2015, $\eta$ is 0.574, and $K^D$ is $6.9 \times 10^6$ ($R^2 = 0.978$), $\vartheta$ is $-0.858$, and $K^I$ is $8.9 \times 10^{16}$ ($R^2 = 0.835$). $\mu$ is the irrigation coefficient, which presents the utilization effectiveness of irrigation water, and its value is defined as 0.487 in 2020 Wang et al. (2017); Wang (2017). Then, three representative crops (wheat, maize and sunflower) are chosen because these crops constitute a large share of the total production in the area. The water demand and virtual water content of these crops are calculated in Table B1.

Water demand in each sector is predicted based on the equations in Section 3.5, namely, $d_1 = 3.50 \times 10^9$, $d_2 = 4.53 \times 10^7$, and $d_3 = 1.94 \times 10^9$. Moreover, the annual water demand in the ecological sector for sediment scouring will to be $1.64 \times 10^8$ m$^3$ in 2020 Wang et al. (2017); Wang (2017).

Predicting regional water demands is necessary to provide references for resource management. This paper presents a modeling approach to measure the domestic, industrial, and agricultural sector water demands in the irrigation district.

## 4 Results

In this study, the disagreement points $dis^U$ and $dis_{i,i=1,2,3}^L$ are set to 33 and 0.3, respectively. The bargaining weights $\alpha_{i,i=1,2,3}$ are defined as 0.3, 0.4, 0.3, respectively. The solving algorithm is coded in MATLAB R2017a.





## 4.1 Crops Inter-regional Exports and International Imports

**Table 2.** Optimal solution to virtual water transfer

| | Agricultural sector | | |
|---|---|---|---|
| | Wheat | Maize | Sunflower |
| $IM_k$ (kg) | $1.30 \times 10^9$ | $3.27 \times 10^8$ | $1.41 \times 10^9$ |
| $EM_k$ (kg) | $7.81 \times 10^8$ | $7.30 \times 10^8$ | $2.08 \times 10^8$ |
| $VW_k \times IM_k$ (m$^3$) | $1.21 \times 10^9$ | $1.08 \times 10^8$ | $2.84 \times 10^9$ |
| $IM_k / y_k$ (hm$^2$) | 242456 | 23630 | 544853 |

Table 2 present the optimal amounts of inter-regional export and international import trade, and corresponding saved irrigation water consumption (virtual water contents in imported crops) and area (virtual land planted if not importing crops). For relatively water-intensive crops, such as sunflower, importing them from other countries is more suitable, in sprite of water utilization efficiency and sectoral vulnerabilities. From $VW_k \times IM_k$, the total virtual water in the imported crops was calculated as $4.16 \times 10^9$ $m^3$, which would save 75.64% of the total available water in the area. The quantification of the land savings cal-

culated by $IM_k / y_k$, is also shown in Table 2. These indicate the quantities of virtual water and land required when substituting imported crops for domestic production. In all, the international import trade affects the water and land savings in the importing area. Hence, to solve the predicted increased water shortages and population in the future, virtual water imports must be added due to inevitable inter-regional export and local consumption.

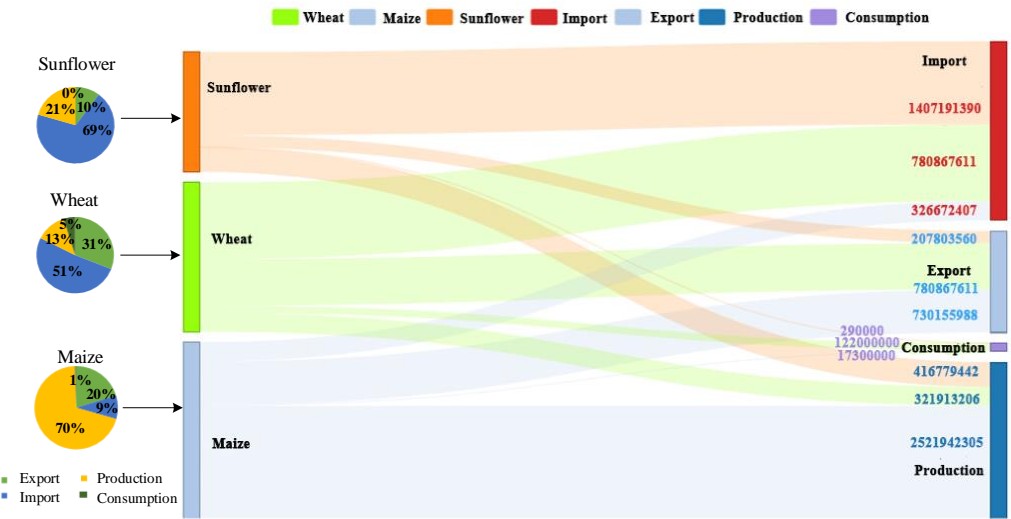

**Figure 5.** The amount of inter-regional export and international import trade

Fig. 5 depicts the proportion of crops in each form (production, import, export and consumption). Wheat, maize and sun-

flower yields are $3.22 \times 10^8$ kg, $2.52 \times 10^9$ kg and $4.17 \times 10^8$ kg, respectively. Crop consumption in this district is far less than





that produced, particularly of sunflowers. Then, the surplus crops can be sold to south China. The area needed to produce the three main crops is calculated at 60,159.45 $hm^2$, 182,424.25 hm$^2$ and 161,373.54 hm$^2$, through a comparison with the national 2020 land-use planning, as shown in Fig. 6, the total land use for crops must be reduced by 16.7% if not allowing international imports. Specifically, to satisfy the requirements for wheat, maize and other food crops, the area for sunflower

decreased significantly. The result indicates that water utilization efficiency improves, especially in land-use planning, through the optimization of the previous planting structure.

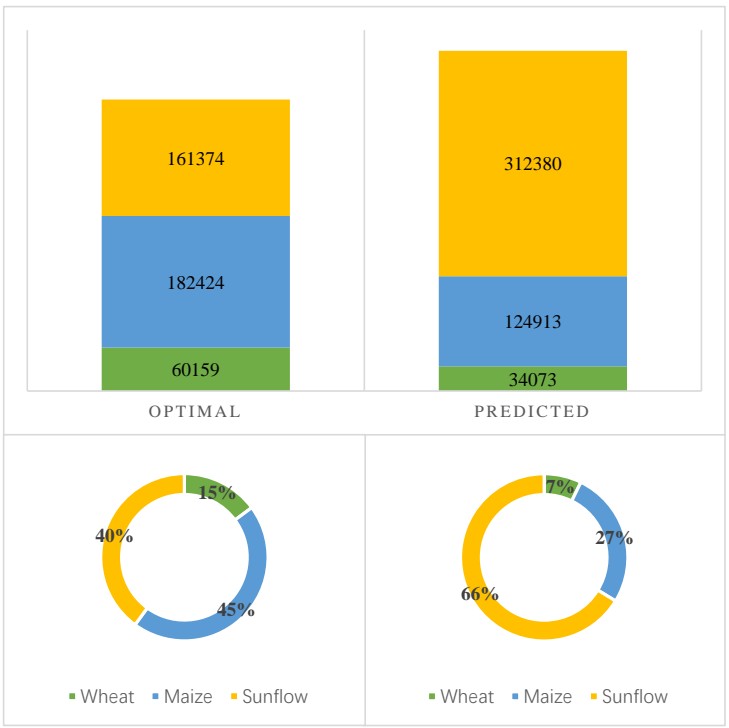

**Figure 6.** Predicted crop areas in planning report compared with optimal value ($hm^2$)

## 4.2 Irrigation industrial and domestic water consumptions accompanied by blue water transfer

Certain differences should be allowed when supplying limited water to sectors. As the model also maximizes the overall water utilization efficiency from the perspective of the system, an equilibrium between the multiple sectors is not always exactly

equal when seeking to satisfy the water demand of each sector.

Table 3 gives an example of a compromised solution for water allocations, transfers, and crop irrigation, with a total amount of 4.45 $\times 10^9$ m$^3$ water is predicted to be consumed in 2020 across the agricultural, industrial, domestic and ecological sectors. More initial water is allocated to the agricultural sector, followed by the domestic sector and the industrial sector, which conforms to actual practices. Through water market (blue and virtual water transaction), blue water is transferred from the





**Table 3.** Optimal water withdrawal results ($m^3$)

| Sectors | Agricultural sector | | | Industrial sector | Domestic sector |
|---|---|---|---|---|---|
| Initial | $X_1$ | | | $X_2$ | $X_3$ |
| water rights | $3.05 \times 10^9$ | | | $1.52 \times 10^7$ | $1.38 \times 10^9$ |
| Water | $X_{11}$ | $X_{12}$ | $X_{13}$ | | |
| irrigation | $3.00 \times 10^8$ | $8.21 \times 10^8$ | $8.41 \times 10^8$ | | |
| Water | | | | WTI | WTD |
| transfer | $-1.09 \times 10^9$ | | | $1.36 \times 10^7$ | $1.07 \times 10^9$ |

agricultural sector to the industrial or domestic sector, $1.36 \times 10^7$ m$^3$ of water is transformed from agricultural sector to industrial sector, $1.07 \times 10^9$ $m^3$ of water is transformed from agricultural sector to domestic sector, which improves the objective functions at the upper and lower levels.

Fig. 7 illustrates the comparisons between water demand, allocation, and consumption, from which a disparity can be seen between the water demand and consumption after blue and virtual water transfers. Fig. 7 (A) shows that the agricultural sector

has the largest gap between water consumption and demand, followed by the industrial sector, while the water demand in the domestic sector is fully satisfied. As shown in Fig. 7 (B), of the major crops, only the water demand for sunflower is not satisfied, of which the planting is replaced by import. Combined with the import results shown in Table 3, our analysis indicates that the reason for the largest gap between water consumption and demand in the agricultural sector is more water-intense crop imports, which can be replaced by import crops. In addition, with more sunflower being imported, the water

utilization efficiency is enhanced, water consumption is saved, and crop land patterns are optimized.

Overall, the above analysis verifies that the proposed model not only helps to optimize the water resource allocation but also leads to land pattern planning based on maximizing the water utilization efficiency and minimizing the vulnerability in each water usage sector. Therefore, it is suggested that when there is insufficient water, blue water can be transferred to the industrial and domestic sectors from the agricultural sectors to enhance the water utilization efficiency and achieve greater

economic benefits; and virtual water import can be used to save irrigation water consumption.

## 5 Discussion

We further explore the reason why not conducting virtual and blue water transfers separately in terms of sectoral water stress values, efficiency, vulnerabilities and so on, and the effects of available water, market price and import price to solve the water allocation problem in the different sectors under future uncertainty.





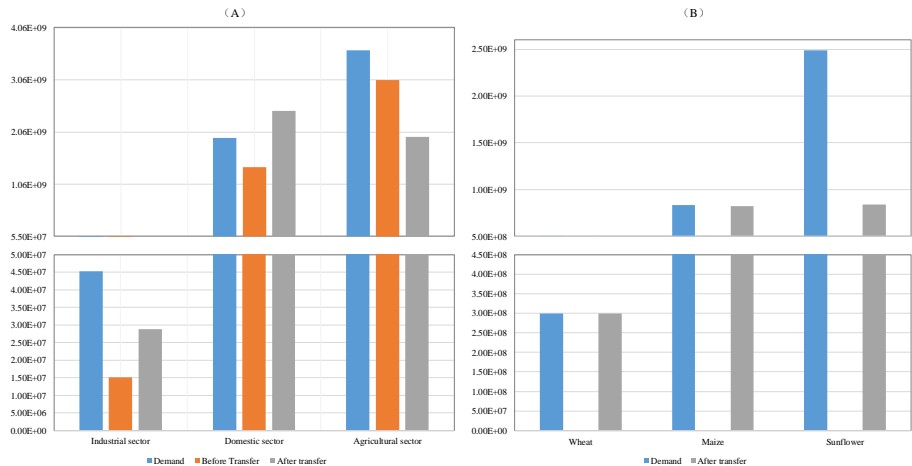

**Figure 7.** Water demand compared with optimal water allocation (m$^3$)

## 5.1 Which sector largely contributes to water stress

Detecting which sector is the largest contributor to regional water stress prior to drawing up a long-term future water plan is of great importance. A blue water scarcity index can be calculated from the production perspective as shown in Eq. (7). $\text{BWS}_1$, $\text{BWS}_2$ and $\text{BWS}_3$ represent the blue water scarcity in the agricultural, industrial and domestic sectors, respectively. The volume of water consumed consequently provides a more accurate basis for estimating water scarcity than the volume of water withdrawal. Hence, $X_2 + \text{WTI}, X_3 + \text{WTD}$, and $\sum_{k=1}^{3} x_{1k}$ are chosen to present the blue water consumption. According to the study of Liu et al. (2017a), the water scarcity of an area is low if the ratio of the water consumption to availability is less than 0.07. The water stress is medium if the ratio is 0.15-0.3 and high if the ratio is greater than 0.3, as shown in Table B2.

$$\text{BWS} = \frac{\sum_{k=1}^{3} x_{1k} + X_2 + \text{WTI} + X_3 + \text{WTD} + X_4}{\text{AW} + \sum_{k=1}^{3} \text{VW}_k \times \text{IM}_k} \tag{7}$$

$$\text{BWS}_1 = \frac{\sum_{k=1}^{3} x_{1k}}{\text{AW} + \sum_{k=1}^{3} \text{VW}_k \times \text{IM}_k} \tag{8}$$

$$\text{BWS}_2 = \frac{X_2 + \text{WTI}}{\text{AW} + \sum_{k=1}^{3} \text{VW}_k \times \text{IM}_k} \tag{9}$$

$$\text{BWS}_3 = \frac{X_3 + \text{WTD}}{\text{AW} + \sum_{k=1}^{3} \text{VW}_k \times \text{IM}_k} \tag{10}$$





The BWS in the Hetao irrigation district was 0.47 after optimization, indicating high water stress. And the BWS values
of agricultural, industrial, and domestic sectors were 0.068, 0.003, and 0.254. Hence, we identify that domestic sector is the
largest contributor to regional water stress, and respectively. Analysis indicates that there are two reasons for these results:
increase in population, more water needed for typical livelihood needs.

## 5.2  The importance of coalitional utilization of blue and virtual water transfers

Virtual water and blue water transfers can have a significantly positive effect on water allocation systems. Previous research
never optimize the water allocation problem incorporating virtual and blue water transfers together; therefore, in the following,
two scenarios are considered to assess the importance of considering them together.

*Scenario A: Comparison to a model without considering virtual water transfers.*

*Scenario B: Comparison to a model without considering blue water transfers.*

The following paragraphs discuss the importance of considering virtual water transfer (down to international imports and
inter-regional exports) and blue water transfers from four perspectives: objective functions, total water consumption, im-
port/export structure, and water stress.

Three compromise solutions are obtained in Table 4. Fig. 8 (A) illustrates the vulnerability and water utilization efficiency
differences after considering blue and virtual water transfers. Overall, blue water transfers improve water utilization efficiency.
The proposed model has the highest water utilization efficiency, followed by Scenario A, with the lowest being Scenario B.
Specifically, the agricultural and industrial sector vulnerabilities are reduced when blue water transfers are included, and the
virtual water transfer is found to have a positive effect on the vulnerability of the industrial and domestic sectors but not on that
of agricultural sectors. Through analysis, we conclude that water transfers enable more water consumption in the sectors that
use less water to produce greater economic benefits, which conforms to the ideas in (Liu and Yang, 2012), i.e., that integrated
water resource management (IWRM) may be a useful approach; such transfers also potentially reduce the degree of crop
self-sufficiency.

A comparison of the optimal allocation of water resources and total consumption under the two scenarios is shown in Fig.
8 (B-C). Compared with Scenario A, the total water consumption of the proposed model decreases from 4,547,691,062 to
4,446,716,331; in detail, the industrial and agricultural sectors receive fewer initial water rights and the domestic sector re-
ceives more initial water rights, and the blue water transfer value declines from 1,126,120,905 to 1,085,970,257. In other
words, even when water evaporation and the complex water pipe network system are considered, the proposed model appro-
priately reduces the water transfer quantities to minimize ecological or economic losses. In comparison, in Scenario B, the
total water consumption solved by the proposed model increases from 4,327,537,703 to 4,446,716,331, with the most water
being consumed by the domestic sectors rather than by the agricultural sectors, which implies that there is a positive effect
propelling the water utilization efficiency and reducing the sector vulnerability when blue water transfers are considered. The
international import and inter-regional export structure also changed, with greater quantities of wheat and sunflower and less
maize being imported; furthermore, there were greater quantities of wheat and less sunflower and maize being exported to





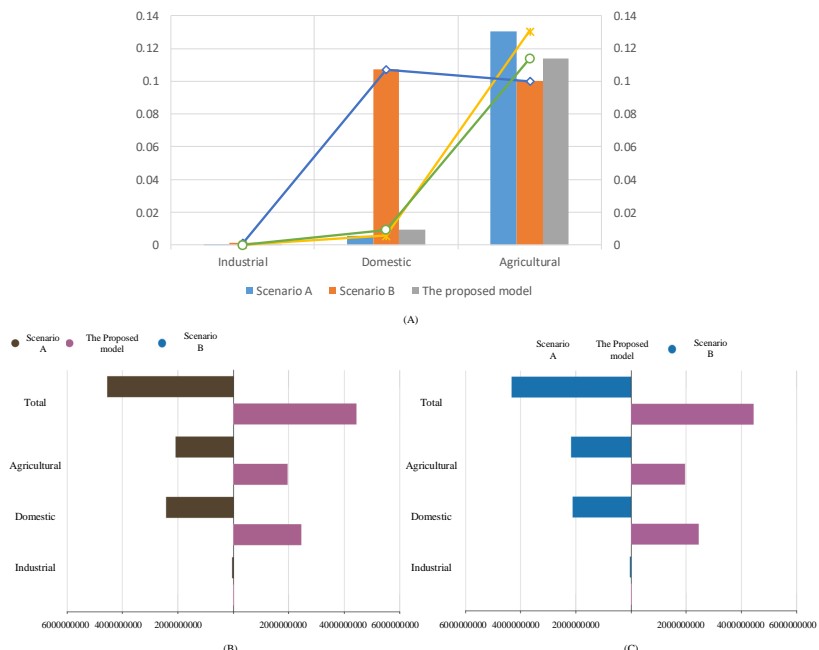

**Figure 8.** Comparisons of the objective values (A), optimal allocation of water resources and total consumption with Scenario A, and Scenario B (B-C)

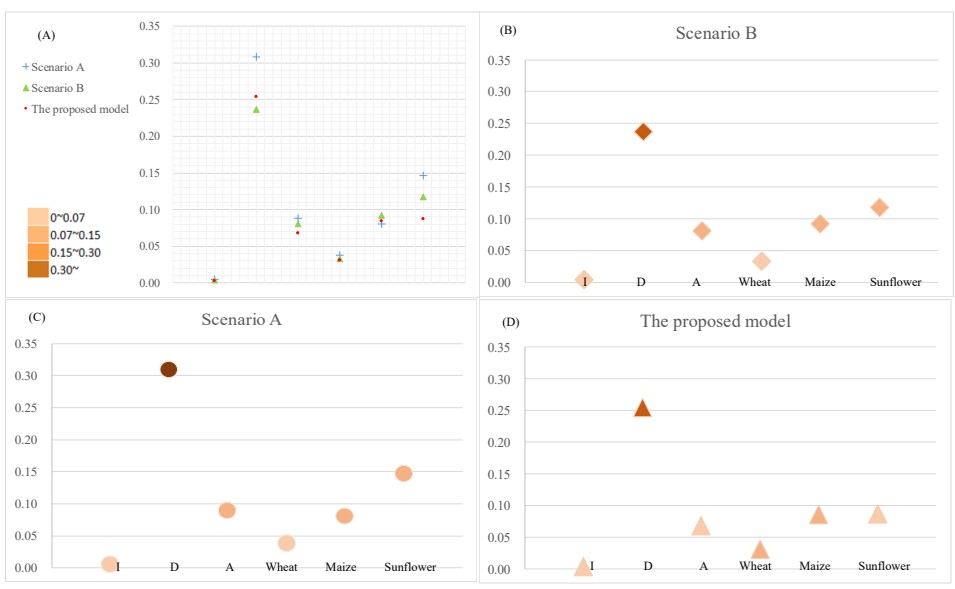

**Figure 9.** Water stress comparisons; (A) Total water stress; (B) Scenario B water stress; (C) Scenario A water stress; (D) Water stress in the proposed model





**Table 4.** Comparison results under different scenarios

|  | Scenario A | Scenario B | Proposed model | Two-stage model |
|---|---|---|---|---|
| Decision variables |  |  |  |  |
| $X_1$ $(m^3)$ | 3207801351 | 2171516527 | 3047229583 | 3082389920 |
| $X_2$ $(m^3)$ | 24211343 | 40419735 | 15189977 | 34315635 |
| $X_3$ $(m^3)$ | 1315678369 | 2115601441 | 1384296772 | 792985984 |
| WTI $(m^3)$ | 18466614 | - | 13552598 | 4493735 |
| WTD $(m^3)$ | 1107654291 | - | 1072417659 | 778055747 |
| $EM_1$ $(kg)$ | - | 260039225 | 780867611 | 115379958 |
| $EM_2$ $(kg)$ | - | 1114440656 | 730155988 | 583203850 |
| $EM_3$ $(kg)$ | - | 632587066 | 207803560 | 456226924 |
| $IM_1$ $(kg)$ | - | 1135941665 | 1297550744 | 482152328 |
| $IM_2$ $(kg)$ | - | 798724570 | 326672407 | 751485047 |
| $IM_3$ $(kg)$ | - | 1042083172 | 1407191390 | 696873618 |
| Objective values |  |  |  |  |
| Lower level |  |  |  |  |
| Agricultural sector | 0.1306 | 0.1002 | 0.1138 | 0.4174 |
| Industrial sector | 0.0002 | 0.0011 | 0.0001 | 0.0080 |
| Domestic sector | 0.0056 | 0.1071 | 0.0092 | 0.0001 |
| Upper level | 71.95 | 66.66 | 74.40 | 111.57 |

neighboring regions. From this paper, therefore, practical managerial insights on crop planting and import/export quantities are derived.

The water stress under the different scenarios is calculated, as shown in Fig. 9. Fig. 9 (A) illustrates the degree to which
water stress is relieved after optimization. The water stress under Scenario A is greater than the stress under the other two scenarios; that is, allowing virtual water transfer can alleviate water stress. The water stress values in the different sectors are then compared, from which it can be seen in Fig. 9 (B-D) that considering blue water transfers can decrease industrial and agricultural sector consumption stress. In addition, virtual water transfers decrease water stress in all sectors. Therefore, virtual water and blue water transfers should be considered in future water management planning.

**5.3   The importance of hierarchical strategic interaction**

One of the widely used models in water resource allocation is a two-stage optimization model. By comparison, in this paper, a Stackelberg game between the leader and followers is fully considered, which displays a strategic interaction before obtaining a compromised solution. To further illustrate the superiority of the proposed model, we compared our work with the two-stage model with the same objective functions for each stage decision maker, and the results are shown in Table 4.
The results indicate that the total consumption solved using the two-stage optimization model is mostly higher than that for the proposed model. In the first stage of the two-stage optimization model, the initial water rights for sector users and virtual water transfers are decided by maximizing the water utilization efficiency; in the second stage, followers decide the qualities of water withdrawal and blue water transfers according to the asymmetric Nash-Harsanyi game model following the decisions in the first stage. By comparing the results in the amount of inter-regional export and international import trade, we find that





the trade scale is squeezed, which is not suitable for the region that sends grains from northern China to southern China. In addition, an increase in the sector vulnerability easily leads to unsustainable development in the irrigation district.

The proposed model includes the strategic interaction, which cannot be ignored by decision makers. Generally, one main reason for the strong discriminating power of the developed model is that it copes with the conflict between the leader and followers and produces a more realistic water allocation strategy.

## 5.4    Sensitive Analysis

The extended Fourier amplitude sensitivity test (EFAST) was used to analyze the sensitivity of the proposed model to crop import prices, available water and water demand.

### 5.4.1    Future uncertainty down to changing available water

As available water is affected by precipitation, in this section, the available water is represented as being random. Then, the
available water is assumed to increase/decrease by 10%, 15% and 20% to assist decision makers in understanding the overall water allocation and utilization situation.

By solving the proposed model using varying available water quantities, optimal results, including the objective values, are obtained, as shown in Fig. 10 and Table 5. Fig. 10 (A) shows that with a decrease in available water, the water utilization efficiency increases; however, when the available water is extremely scarce, the water utilization efficiency decreases sharply.
During the experimental studies, the fluctuations in the industrial usage sector are the smallest, followed by those of the agricultural usage sector, with the fluctuations of the domestic usage sector being the largest, indicating that the domestic usage sector is more sensitive to water availability. Based on the average vulnerability values in each sector shown in Fig. 10 (B), the vulnerability of the industrial usage sector is the least sensitive to available water, followed by that of the agricultural usage sector, with the domestic usage sector being the most sensitive. The water stress in the different sectors is compared in
Fig. 10 (C). Combined with Fig. 10 (D), it can be seen that when the available water decreased by 10%, the water-allocation structure changed slightly, with the water stress in each sector increasing compared to that in the proposed model. When the available water decreased by 15%, the total water consumption decreased, and the import trade (wheat, maize and sunflower) increased, which resulted in an overall decrease in the water stress values.  In contrast, when the available water increased by 10%, the total water consumption decreased, and the water stress in the agricultural and domestic usage sectors decreased due
to a greater precipitation supplement. When the available water increased by 15%, more water was allocated to the domestic and agricultural sectors, leading to an expansion in the export trade quantities. Two extreme scenarios with lower water utilization efficiencies are found in the seven scenarios. When the available water decreases by 20%, more water is needed in the agricultural sector because there is less precipitation during the crop growing periods. As a result, significantly less water is allocated to the domestic sector, and more water is made available for crop irrigation. The other extreme scenario is when the
available water increases by 20%; in this scenario, the total water consumption largely decreases due to disasters (e.g., floods) that might occur, and the virtual water transfer decreases because of the extreme environmental conditions.





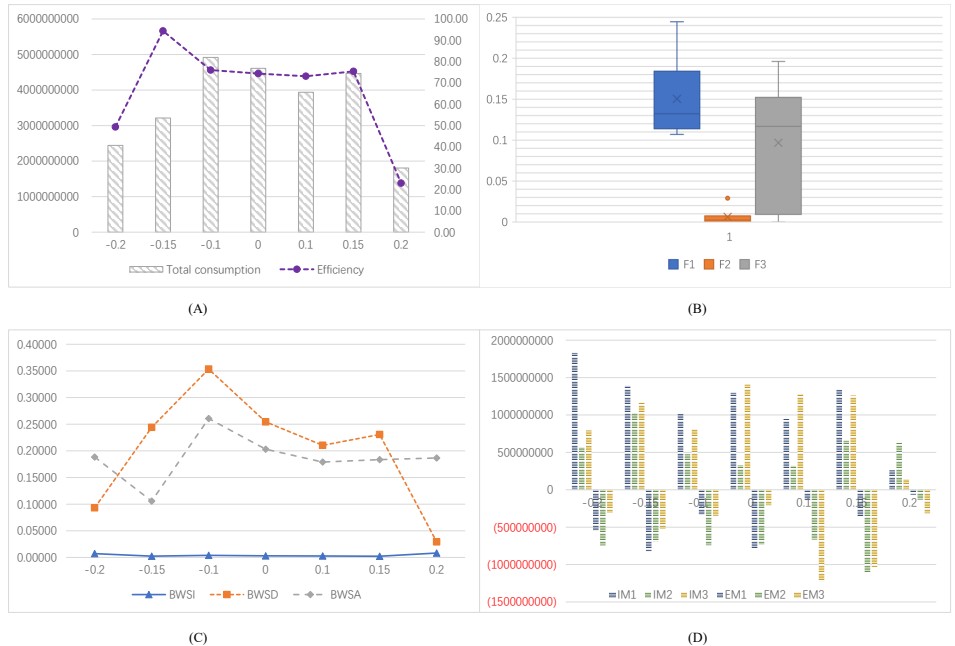

**Figure 10.** Sensitivity analysis of available water (A) Total consumption and water utilization efficiency under different available water; (B) Sector's vulnerability; (C) Water stress in different sectors; (D) Virtual water transfers.

**Table 5.** Optimal solutions under varying available water constraints

|        | -20%       | -15%       | -10%       | 0          | 10%        | 15%        | 20%        |
|--------|-----------|-----------|-----------|-----------|-----------|-----------|-----------|
| $X_1$  | 1880616733 | 1838816689 | 3254046602 | 3047229583 | 2592206880 | 3074860767 | 1503875715 |
| $X_2$  | 29115294   | 17542527   | 20013408   | 15189977   | 19057423   | 13633183   | 31069166   |
| $X_3$  | 369525207  | 1192017093 | 1480915742 | 1384296772 | 1161597722 | 1215323490 | 106882843  |
| WTI    | 27691014   | 3454404    | 10721323   | 13552598   | 7669815    | 9925616    | 29739036   |
| WTD    | 365711343  | 920355891  | 1237891631 | 1072417659 | 862496257  | 1168282579 | 106425315  |
| $EM_1$ | 541839827  | 820390947  | 330257166  | 780867611  | 133708047  | 359419705  | 67383848   |
| $EM_2$ | 746332995  | 697557298  | 742943057  | 730155988  | 666548137  | 1102868468 | 131628478  |
| $EM_3$ | 306044870  | 521941328  | 346936514  | 207803560  | 1226163125 | 1033216002 | 315440259  |
| $IM_1$ | 1829968780 | 1388283830 | 1013412668 | 1297550744 | 946557516  | 1338741252 | 261184655  |
| $IM_2$ | 564043019  | 1032461308 | 475691412  | 326672407  | 319629355  | 655693017  | 626996647  |
| $IM_3$ | 794636883  | 1162988995 | 813745354  | 1407191390 | 1278708716 | 1260301105 | 137210370  |

Therefore, it is concluded that extremely dry or wet environments are not beneficial to water utilization efficiency, and the virtual water transfer is sensitive to the available water; with a decrease in available water, the import trade increases. With an increase in available water, there is an increase in the export trade and particularly in sunflower exports (a crop that brings higher economic benefits with greater water consumption). However, in extreme environments, the balance between production and living can be destroyed. To ensure stable development in this case, close management by decision makers is necessary to ensure food self-sufficiency with a lower virtual water transfer. Therefore, because extreme environments can affect water





utilization efficiency, suitable measures are necessary for future development. For example, in a wet environment, crops that have greater virtual water can be planted and exported. In a dry environment, it is suggested that crops that require more
irrigation be obtained through import trading.

### 5.4.2   Future uncertainty down to sectoral water demand

Using the proposed model, the water demand in each sector is assumed to decrease by 10% and 20% by policies implemented to encourage water-conservation irrigation techniques, improve resident water conservation awareness, construct water reuse schemes, manage water use mechanisms, and expand water conservation investment. Thus, 7 scenarios are analyzed, including one that had no change (the result from the proposed model).

**Table 6.** Objective values and total consumption for the 10% and 20% reductions in water demand

|  |  | Agricultural (-10%) | Industrial (-10%) | Domestic (-10%) |
|---|---|---|---|---|
| $F_1$ | 0.00006 | 0.00030 | 0.00474 | 0.00162 |
| $F_2$ | 0.00920 | 0.04632 | 0.15199 | 0.02506 |
| $F_3$ | 0.11382 | 0.26058 | 0.14329 | 0.13170 |
| Efficiency | 74.40 | 95.42 | 77.48 | 57.68 |
| Total consumption | 4610716332 | 3904770411 | 4069341955 | 3018694351 |
|  |  | Agricultural (-20%) | Industrial (-20%) | Domestic (-20%) |
| $F_1$ | 0.00006 | 0.00002 | 0.00030 | 0.00460 |
| $F_2$ | 0.00920 | 0.06672 | 0.03920 | 0.03719 |
| $F_3$ | 0.11382 | 0.11124 | 0.11939 | 0.11469 |
| Efficiency | 74.40 | 59.99 | 72.24 | 38.93 |
| Total consumption | 4610716332 | 3088935079 | 4225900802 | 2929174334 |


    To monitor the performance and assess actions relevant to each scenario, a basic assessment paradigm is first constructed to track and develop appropriate policies. Within the assessment paradigm, technology, socioeconomic development, resource usage and vulnerability are combined. A "Total consumption" index is applied to reflect the natural resources and ecosystems. A "Water utilization efficiency" is applied to reflect socioeconomic development. A "vulnerability" variable is included to
reflect the performance in each sector, and "Investment and risk management" is included as an external influencing factor that reflect the awareness and support levels. Based on the solution results, potential policies are analyzed for the water usage sectors. Referring to the relative results presented in Table 6, we analyze the results from the perspective of different water usage sectors as follows.

    With a decrease in agricultural water demand, water utilization efficiency improves, and total consumption decreases. How-
ever, if there is a very large reduction in water demand, industrial sector vulnerability would be aggravated. Therefore, healthy water conservation policies are suggested, such as increasing crop yields that use less water without increasing the burden on the other sectors to ensure sustainable development. For example, modifying irrigation frequency and adopting a single irrigation strategy are operative policies that can reduce the amount of irrigation while achieving acceptable crop yields. Finally, a balance between the need for food self-sufficiency and water conservation should also be considered when virtual water





imports are being considered to reduce irrigation in the agricultural sector; however, excessive imports are also harmful to farmers. Therefore, import quotas need to be modified in line with the changing environment.

When industrial water demand decreases by 10%, overall efficiency is found to improve; however, too great of a reduction could result in a negative influence on the sustainability of a water allocation system. The research by (Zou and Liu, 2014)found that there was an "inverse U-shaped relationship" between industrial water demand and industrial added value and the industrial
water recycling rate; that is, a reduction in the industrial water consumption does not always increase the industrial added value. In other words, water conservation in the industrial sector has a positive effect on economic development only after the industrial added value/water recycling rate increases to a certain value, which means that reductions in industrial water demands should be accompanied by industrial structural adjustments and industrial water-recycling rate improvements, both of which require capital investment and government support. Overall, some reductions in industrial sector water use can be
beneficial; however, too great of a reduction could adversely affect the sectors' economic development and increase system vulnerability.

It can be seen that blindly decreasing domestic water demands can harm efficiency and increase vulnerability; however, it can reduce total consumption in the water allocation system, as shown in Table 6. Generally, because there is only low technology support for domestic water supplies, any reduction in domestic water demand can be harmful to economic development;
therefore, a better approach is to develop incentives to encourage households to reduce their domestic water consumption, such as purchasing water-saving devices and encouraging environmentally friendly behaviors (e.g., preventing animal waste from entering waterways and putting rubbish in bins). A survey suggested that "73.0% of respondents agreed 'Water conservation actions by householders can significantly reduce the amount of water used in urban areas'".

Overall, the sensitivity analysis provides insights into future investments in water conservation and sustainable develop-
ment. The suggested strategies for the agricultural sector include modifying irrigation frequency, adopting a single irrigation strategy, modifying plant structures to fit with water demand reductions and limiting import quotas in each area based on the changing environmental conditions. The suggested strategies for the industrial sector include increasing capital investment and government support prior to increasing industrial water demand. The suggested strategies for the domestic sector include implementing incentive policies to encourage households to reduce domestic water consumption rather than limiting production.
Therefore, several policies can be implemented to guarantee sustainable water utilization and reduce total consumption.

### 5.4.3 Future uncertainty down to import crop prices

Varying prices of import crops are input into the model to analyze the effect of import prices on water allocation, withdrawal and transfer processes within the system. Then, each crop is assumed to increase/decrease by 10% to show the feasibility of the proposed model.
Table 7 presents the optimal solution for water allocation, blue water transfers and inter-regional exports and international imports. First, it is found that the fluctuation of results under different import prices is smaller than that under varying available water and sectoral water demands. Second, Fig. 11 shows that the model is more sensitive to water price variation in maize,





followed by wheat and sunflower, where the conclusion can be derived from the total consumption and qualities of inter-regional exports and international imports.

**Table 7.** Optimal solutions under varying import crops' prices

|  | No change | Wheat import price | | Maize import price | | Sunflower import price | |
| --- | --- | --- | --- | --- | --- | --- | --- |
|  | 0 | -10% | 10% | -10% | 10% | -10% | 10% |
| $X_1$ | 3047229583 | 2783032608 | 3195815495 | 2687606880 | 3183528493 | 2918541967 | 3160649954 |
| $X_2$ | 15189977 | 14935370 | 14753100 | 19637270 | 14137836 | 19969847 | 16945271 |
| $X_3$ | 1384296772 | 1376800804 | 1247364753 | 1215323490 | 982807458 | 1381203589 | 1288879651 |
| WTI | 13552598 | 14644536 | 11986419 | 9292630 | 11021744 | 10871633 | 10573207 |
| WTD | 1072417659 | 1033736303 | 951952634 | 1054795267 | 870739733 | 967270913 | 832218168 |
| $EM_1$ | 780867611 | 863861648 | 530949201 | 592783566 | 956388371 | 568247308 | 555952983 |
| $EM_2$ | 730155988 | 711586152 | 748180306 | 422095094 | 343021534 | 1070282648 | 687931254 |
| $EM_3$ | 207803560 | 192591875 | 198838918 | 588329447 | 212332515 | 511504901 | 799972771 |
| $IM_1$ | 1297550744 | 1451754404 | 1227933488 | 1244164347 | 1546075857 | 1378169935 | 1314312083 |
| $IM_2$ | 326672407 | 328894104 | 354435544 | 446089695 | 314383237 | 353592654 | 326816881 |
| $IM_3$ | 1407191390 | 1266494343 | 1476080837 | 1404828996 | 144602917 | 1430439473 | 1386190486 |

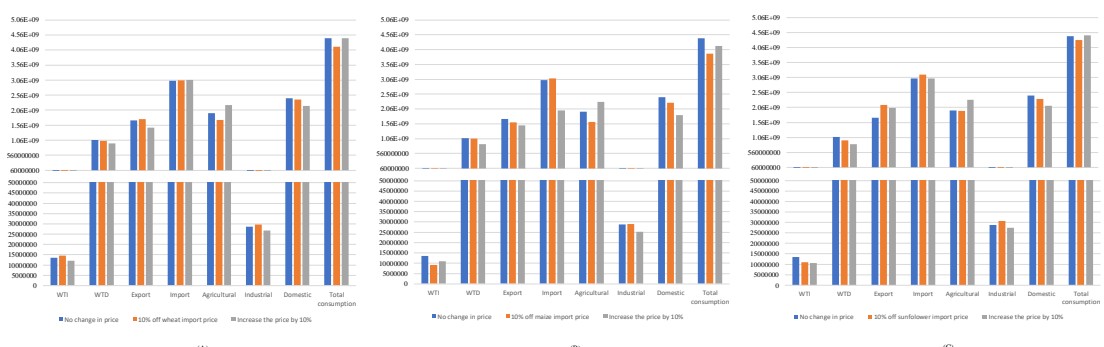

**Figure 11.** Values under different import prices (A) Results change caused by different wheat import price; (B) Results change caused by different maize import price; (C) Results change caused by different sunflower import price.

### 5.4.4  Future uncertainty down to market prices

With the implication of solving model, we define the parameter $\theta$ as 0.5, and obtained an optimal solution to water withdrawal and transfers. We probe deeper by conducting a sensitive analysis to determine the implications of differences in decision variables.

Table 8 present the results of optimized decision variables and Fig. 12 present the variation in (A) agricultural water with-drawal; (B) Industrial water withdrawal; (C) Domestic water withdrawal; (D) Water transfer from agricultural sector to indus-trial sector; (E) Water transfer from agricultural sector to domestic sector; (F) Total water consumptions. The fitted line in Fig. 12 is a 4rd-degree polynomial. The six smoothing lines find that there is a fluctuation with the increase of $\theta$ (in other words,





with the decrease of market price). The results imply that the price elasticity of water transaction is not linear, and the specific relationship can be defined in the future.

**Table 8.** Optimal solutions under varying water market prices

|        | $\theta = 0.15$ | $\theta = 0.30$ | $\theta = 0.50$ | $\theta = 0.65$ | $\theta = 0.80$ |
|--------|-----------------|-----------------|-----------------|-----------------|-----------------|
| $X_1$  | 2605944850      | 3419244827      | 3047229583      | 2371527708      | 2436421714      |
| $X_2$  | 19057423        | 21264259        | 15189977        | 31636639        | 31707489        |
| $X_3$  | 1161597722      | 1143784241      | 1384296772      | 742974238       | 691174164       |
| WTI    | 7669815         | 16923705        | 13552598        | 23681752        | 29827397        |
| WTD    | 862496257       | 1113539231      | 1072417659      | 572148734       | 657717900       |
| $EM_1$ | 133708047       | 150916369       | 780867611       | 233785058       | 301852668       |
| $EM_2$ | 666548137       | 938686452       | 730155988       | 1140060732      | 507944355       |
| $EM_3$ | 1226163125      | 766444693       | 207803560       | 1038707269      | 206333010       |
| $IM_1$ | 946557516       | 825779180       | 1297550744      | 537365650       | 527116977       |
| $IM_2$ | 319629355       | 718937991       | 326672407       | 570713541       | 1331964766      |
| $IM_3$ | 1278708716      | 990772615       | 1407191390      | 1409524883      | 1068773509      |

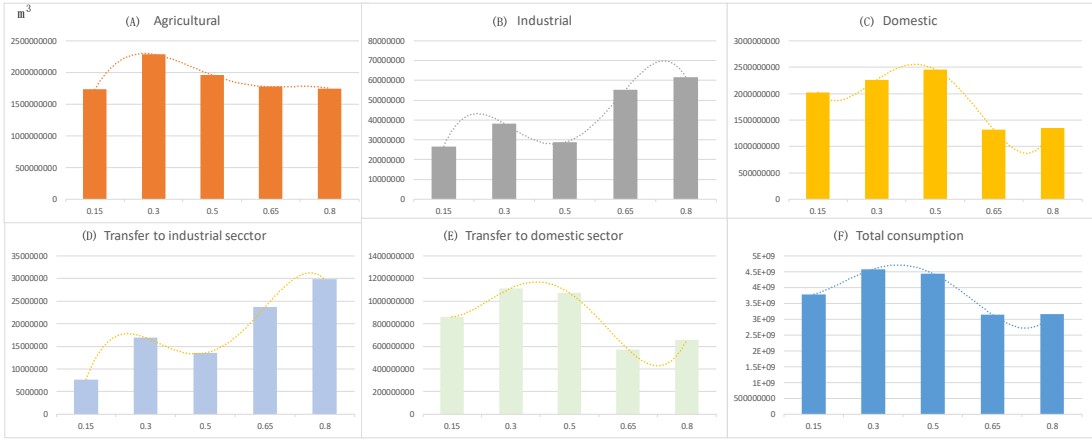

**Figure 12.** Values under different market price (A) agricultural water withdrawal; (B) Industrial water withdrawal; (C) Domestic water withdrawal; (D) Water transfer from agricultural sector to industrial sector; (E) Water transfer from agricultural sector to domestic sector; (F) Total water consumptions.

# 6   Conclusions

In this study, we proposed a novel model based on Stackelberg-Nash-Harsanyi game theory for analyzing the water reallocation problem in the promise of water transfers and crops transactions. To describe realistic water allocation, withdrawal and transaction processes, this paper employed a bilevel framework with one leader and multiple followers, where a water affairs bureau was in the leadership position and multiple water usage sectors were in a lower position. From the water consumption


side, four water-usage sectors (agricultural, domestic, industrial and ecological) were considered, with the initial water rights given to the ecological sector being pre-determined by government planning. As populations increase, the water withdrawal competition between the sectors becomes increasingly serious. To decrease the imbalance between water demand and water supply, as well as to promote sustainability in different sectors, vulnerability, including the destroying degree (caused by deficient water withdrawal) and economic loss (caused by excess water withdrawal), were the lower level objective; to allocate the
limited water resources, the water utilization efficiency was maximized at the upper level across the whole system.

Blue water transfer and virtual water transfer are examples of mechanisms that essentially relieve uneven distribution of water. In this study, one incorporated idea is that the blue water can be reallocated to industrial and domestic sectors to develop water-saving agriculture. The other incorporated idea is that the virtual water existing in crops is quantified to optimize inter-regional exports and international imports. Virtual water transfer path is from regions defined by comparative advantage of water resources to those defined by comparative disadvantage of water resources so that water utilization efficiency was
improved for the regions involved. Besides, blue water transfer path is from sectors with lower water vulnerability value to those with higher water vulnerability value so that vulnerability caused by exceed/lack the quantity of water demand decreased for the sectors involved. All told, having incorporated the concepts of blue and virtual water transfers, our model is able to further relieve the water scarcity stress as well as reduce the vulnerability and increase the water utilization efficiency. During
the dynamic strategic interaction, the leader modifies its decision based on the best-response strategies by the followers.

To verify the feasibility and practicality of the developed model, a real-world application was conducted in the Hetao irrigation district. The results found that water demand in the domestic sector was first satisfied, followed by that of the agricultural and industrial sectors; blue water transfer provided an opportunity for each sector to achieve an efficient utilization of water, and virtual water transfer provided a new opportunity for water conservation and land saving. To be specific, some initial wa-
ter rights could be transferred from the agricultural sector to the industrial and domestic sectors, and key crops, particularly water-intensive crops (e.g., wheat and sunflower), could be imported from other countries rather than being grown domestically. After the blue water transfers, the domestic water demand can be initially satisfied, which conforms to the principles of sustainable development. Furthermore, to demonstrate the superiority of the developed model, two comparative scenarios were considered: one without virtual water transfers and the other limiting blue water transfers. The analysis of Scenario A
highlighted the importance of virtual water transfers in significantly alleviating water usage stress. Furthermore, the analysis of Scenario B verified that blue water transfers reduced vulnerability in each sector and demonstrated how inter-regional exports, international imports and blue water transfers can result in higher water utilization efficiencies in all three sectors.

Considering the changing environment and water-saving strategies, several scenarios were then assessed, including one without any change (the results of the proposed model). Faced with varying water availability, it was concluded that in wet
environments, crops with more virtual water could be planted and exported. In contrast, in dry environments, crops that require more irrigation should be imported. An analysis of how to reduce water demand in different water usage sectors is discussed. It is worth noting that blindly decreasing domestic water demands can harm efficiency and increase vulnerability; however, it can reduce total consumption in the water allocation system. When industrial water demand decreased by 10%, the overall efficiency showed improvement; however, too great of a reduction could result in a negative influence on the sustainability





of the water allocation system. With a decrease in agricultural water demand, water utilization efficiency improved, and total consumption decreased. However, if there is a very large reduction, the industrial sector vulnerability would be aggravated. Hence, several policies can be implemented in terms of various sectors to alleviate the regional water stress and the negative impact caused by the crop trade, as well as to safeguard regional food security to develop a sustainable irrigation district.

Under the changing market in terms to crop transaction price and water transfer price, several scenarios were conducted to 535 probe deeper the optimal allocation and transfers strategies. The results denoted that the model was more sensitive to available water and sectoral demand, instead of crop import price. Further, the model was more sensitive to water price variation in maize, followed by wheat and sunflowers. Then the fitted smoothing lines based on the solved values implied that the price elasticity of water transaction is not linear, which gave a direction for formulation demand function in future research.

This work focused on water allocation in a bilevel framework. Given that we explored only one leader and multiple followers, 540 there are possibilities for multiple leaders and multiple followers in future research. Furthermore, a dynamic analysis from the time perspective may be interesting.

## Appendix A: Notations

The following notations are used to develop the model.

**Indexes:**

$k$ : crop indicators, $k = 1$ for wheat, $k = 2$ for maize, $k = 3$ for sunflower

$i$ : water usage indicator, $i = 1$ for agricultural sector, $i = 2$ for industrial sector, $i = 3$ for domestic sector, $i = 4$ for ecological sectors

**Parameters:**

$p_i$ : water price set by the leader for sectoral water uses, RMB/$m^3$

$\text{ERW}_i$ : economic return per unit of water consumption in sectors, $i = 1, 2, 3$, RMB/m$^3$

$\text{ERP}_k$ : economic return from agricultural products exports, $i = 1$, where $\text{ERP}_k = \text{ERW}_k \times VW_k$, RMB/$m^3$

$c_k$ : economic costs because of agricultural products imports RMB/kg

TC : transaction cost per unit of water resource from agricultural sector to industrial or domestic sectors RMB/m$^3$

$\mu$ : the irrigation coefficient, which presents the utilization effectiveness of irrigation water

$A$ : total available area for crop planting, hm$^2$

$\phi_{\text{ind}}$ : the gross industrial output value, RMB

$R_k$ : effective rainfall, mm

$\varpi_k$ : crop $k$ consumption per unit in the Hetao irrigation area, kg/person

**Auxiliary variables (continuous variable)**

AW : maximum volume of available water in Hetao irrigation district, m$^3$

PTI : price of water transfers to industrial sector, RMB/m$^3$

PTD : price of water transfers to domestic sector, RMB/m$^3$





$w_k$ : water irrigation for crop $k$, /m$^3$

$W_k$ represents the blue and green water components in crop $k$, m$^3$

$y_k$ represents the crop yield per unit of irrigation area, kg/hm$^2$

$l_k$ : total yield of crop $k$, $kg$

VW$_k$ : virtual water content of crop $k$, m$^3$ kg$^{-1}$

$A_{1k}$ : area allocated to crop $k$, hm$^2$

$d_i$ : water demand of sectors, $i = 1, 2, 3, 4$, m$^3$

$d_{1k}$ : water demand of crops in agricultural sector, $k = 1, 2, 3$, m$^3$

$\phi_{\text{pop}}$ : per capita disposable income, RMB

POP: the population in the Hetao irrigation area

**Decision variables**

$X_i$ : initial water rights in sectors, $i = 1, 2, 3, 4$, determined by the upper-level decision maker, m$^3$ (continuous variable)

EM$_k$: quantity of products exports, determined by the upper-level decision maker, kg (continuous variable)

IM$_k$: quantity of products imports from international trade, determined by the upper-level decision maker, kg (continuous variable)

WTI : water transfer from agricultural sector to industrial sector, determined by the lower-level decision makers, m$^3$ (continuous variable)

WTD : water transfer from agricultural sector to domestic sector, determined by the lower-level decision makers, m$^3$ (continuous variable)

$x_{1k}$ : water irrigated to crop $k$ in the agricultural sector, determined by the lower-level decision maker, $\sum_{k=1}^{3} x_{1k} = X_1$, m$^3$ (continuous variable)

**Appendix B:**

**Table B1.** Water demand and Virtual Water Content in 2020

|  | Wheat | Maize | Sunflower |
|---|---|---|---|
| $W_k$(m$^3$/hm$^2$) | 4980 | 4500 | 5210 |
| $ET_k$(mm) | 498.0 | 450.0 | 521.0 |
| $R_k$(mm) | 71.0 | 125.2 | 134.5 |
| $w_k$(mm) | 876.80 | 666.94 | 793.63 |
| $y_k$ (kg/hm$^2$) | 5351.7 | 13824.6 | 2582.7 |
| $VW_k$(m$^3$/kg) | 0.93 | 0.33 | 2.02 |





**Table B2.** Category of blue water stress in the Hetao irrigation district

| Category | Values of blue water stress |
|---|---|
| low water stress | <0.07 |
| low to medium water stress | 0.07-0.15 |
| media water stress | 0.15-0.30 |
| high water stress | >0.30 |

**Table B3.** Pseudo code for the proposed model.

**Input:** The correlation parameter, disagreement points ($dis^U$, $dis_i^L$), bargaining weights ($\alpha_i$) ;

**Output:** A satisfactory solution and objective functions for the proposed model;

**Step 1:** Initialize population, construct fitness functions for the upper objective (Eff) and an auxiliary fitness function for the Nash-Harsanyi model in the lower level ( $\prod\limits_{i=1,2,3} \left( F_i - dis_i^L \right)^{\alpha_i}$ );

**Step 2:** Randomly generate an initial solution, maximize the fitness function in the lower level

 IF (any lower level objective functions are lower than their disagreement points)

 {generation++;

 population [generation+1]=Select (Mutate(Crossover (parents)))};

 Else: go to step (3);

**Step 3:** Input the satisfactory solution from step (2), maximize the fitness function in the upper level

 IF (upper level objective function is lower than its disagreement point)

 {generation++;

 population [generation+1]=select (mutate(crossover (parents)))};

 Else: keep the current decision variables and corresponding solutions and go to step (4);

**Step 4:** Feedback

 IF(the updated lower level objective functions are still better than the disagreement points)

 {IF (generation > Maximum generation )

  {Output all the decision variables

  End loop}}

 Else: go to step (2)

**Appendix C:**

**C1 Virtual water content**

This study provides detailed quantitative information on crop imports and exports. The principle for assessing the virtual water content (denoted by $\mathrm{VW}_k, k = 1, 2, 3$, $\mathrm{m^3\ kg^{-1}}$ ) of a food was proposed by the Food and Agriculture Organization (FAO) as the amount of water per unit of food that is consumed during the production process Zeng et al. (2012), as shown in Eq. (C1).

$$\mathrm{VW}_k = \frac{W_k}{y_k} = \frac{10(\mu \times w_k + R_k)}{y_k}, \tag{C1}$$

where effective rainfall is denoted as $R_k$ (mm) and irrigated water is denoted as $w_k$ (mm). $\mu$ is the irrigated coefficient, which presents the utilization effectiveness of irrigation water. $W_k$ ($\mathrm{m^3}$), calculated by $10(\mu \times w_k + R_k)$, presents the water components in crop $k$, which consist of effective irrigated water and rainfall. The value of 10 is used to convert mm to $\mathrm{m^3 hm^{-2}}$. $y_k$ ($\mathrm{kg\ hm^{-2}}$) represents the crop yield per irrigation area unit. As rainfall is an additional supplement for crop water demands, 595 in addition to evapotranspiration (ET), water irrigation quantities are somewhat influenced by the effective rainfall in different





**Table B4.** Overview of input variables, data sources and determination

| Input variable | Source | Determination |
|---|---|---|
| Water demand in each sector | BayanNur Water Resources Bulletin | Predicted by Eqs. (C6)-(C9) |
| Output in each sector per unit of water | Published paper Liu (2016) | |
| Consumption of each crop per capita | Bayna Noaoer yearbook | Average value from years 2012-2015 |
| Irrigation coefficient | Published paper Wang (2017) | – |
| The price of water | Published paper Wang (2017) | – |
| Prices of imported agricultural crops | Wind database, FAO database. | Calculated by average annual import price of each crop based on data for the period 2012-2015 |
| Consumptions of each crop yearly | Bayna Noaoer yearbook, Bayannur water resources bulletin | Calculated by $\sum_{k=1}^{3} \text{POP}_k \varpi_k$ |
| Prices of blue water transfers | Published paper: Erfani et al. (2014) | Calculated by Eq. (1) |
| Transfer cost | Website China water exchange (http://cwex.org.cn/lising/) | Average value |
| The hydrologic data | Inner Mongolia Statistical Yearbook, Bayna Noaoer yearbook, China Meteorological Data Sharing Service System, Published paper Wang (2017) | – |
| Agricultural data | Hetao Irrigation District Agricultural Statistical Data, Published paper Wang (2017) | – |

seasons in addition to evapotranspiration (ET), as shown below.

$$ET_k = kET_0, \tag{C2}$$

$$ET_0 = \frac{0.408\Delta \left(G_n - G\right) + \gamma \frac{900}{T+274}\kappa \left(e_s - e_a\right)}{\Delta\gamma \left(1 + 0.34\kappa\right)}, \tag{C3}$$

$$w_k = \frac{\left(ET_k - R_k\right)}{\mu}, \tag{C4}$$

$$R_k = \beta P_k. \tag{C5}$$

The accumulated ET over the crop-growing period, shown in Eq. (C2), was developed by the FAO (Su et al., 2014), where $\Delta$ is the slope of the vapor pressure curve (kPa °C$^{-1}$), $G_n$ is the net radiation at the crop surface (MJm$^{-2}$ day$^{-1}$), $\gamma$ is the 600 psychrometric constant (kPa °C$^{-1}$), $T$ is the average air temperature (°C), $\kappa$ is the wind speed measured at 2 m height (ms$^{-1}$), $e_s$ is the saturation vapor pressure (kPa), and $e_a$ is the actual vapor pressure (kPa) (Su et al., 2014).





## C2 Agricultural Water Demand

Agricultural water demand is the sum of the crops' demand for water, as calculated in Eqs. (C6) and (C7), for which several representative crops were chosen.

$$d_{1k} = w_k A_k,$$
(C6)

$$d_1 = \sum_{k=1}^{3} d_{1k},$$
(C7)

where $d_{1k}$ is the water demand of crops in the agricultural sector, and $A_k$ is the area allocated to crop $k$.

## C3 Domestic Water Demand

Domestic water demand includes all water consumed in a given period for all residential purposes, e.g., in-house water use for
kitchens, laundry and baths, and outside uses in gardens. The domestic water demand can be determined based on population and income growth projections (Cai, 2002; Young and Haveman, 1985). From many studies on water demand, it was concluded that per capita disposable income was a key influence on water demand; therefore, a power function between water demand and water price was determined.

$$d_3 = K^D \phi_{pop}^{\eta},$$
(C8)

where $K^D$ is a constant, $\phi_{pop}$ is the per capita disposable income, and $\eta$ is the demand income elasticity coefficient.

## C4 Industrial Water Demand

The projection of industrial water demand depends upon income (gross industrial output value). Through the data obtained from the BayanNur Water Resources Bulletin, future industrial water demand is projected as follows:

$$d_2 = K^I \phi_{ind}^{\vartheta}$$
(C9)

where $K^I$ is the constant, $\phi_{ind}$ is the gross industrial output value, and $\vartheta$ is the demand income elasticity coefficient.

## C5 Ecological Water Demand

The water demands for environmental protection and ecological systems are assumed to be constant at the present stage of the study, and these values are determined by the local government based on the consideration of current water use and climate change.





## Appendix D: Model proposal

### D1 Objective Functions

To ensure both economic benefits and water demand satisfaction in different sectors, the water affairs bureau seeks to maximize the water resource system water utilization efficiency, which is denoted by Eq. (D1). In addition, the leader needs to determine the crop quantities that can be exported or imported to achieve economic development. The followers need to independently minimize their own vulnerabilities. In the following, the mathematical objectives and constraints are described, in which index $i$ belongs to the set of all water users; agricultural, industrial, and domestic sectors; and index $k$ distinguishes the different crop types.

Because of the shortage of water resources, the decision maker focuses more on water utilization efficiency. The fractional linear programming (D1) reflects the utilization efficiency index, in which the import and water transfer costs are considered when determining the total economic benefits.

$$\text{maxEff} = \frac{\text{Re}}{\text{Cons}} \tag{D1}$$

$$\text{Re} = \text{Income} - \text{Cost},$$

$$= \left( \sum_{k=1}^{3} (\text{ERP}_k \times \text{EM}_k) + \text{WTI} \times \text{PTI} + \text{WTD} \times \text{PTD} - \sum_{k=1}^{3} (p_1 \times x_{1k}) \right) + (X_2 \times \text{ERW}_2 +$$

$$\text{WTI} \times \text{ERW}_2 - \text{PTI} \times \text{WTI} - p_2 \times X_2) + X_3 \times \text{ERW}_3 + \text{WTD} \times \text{ERW}_3 - \text{PTI} \times \text{WTD}$$

$$-p_3 \times X_3 - \sum_{k=1}^{3} (c_k \text{IM}_k) - \text{TC} \times (\text{WTI} + \text{WTD}), \tag{D2}$$

$$\text{Cons} = \left( \sum_{k=1}^{3} x_{1k} + \text{WTI} + \text{WTD} \right) + X_2 + X_3, \tag{D3}$$

where Re is the total economic returns in the agricultural, industrial and domestic sectors, as shown in Model (D2). $\left( \sum_{k=1}^{3} (\text{ERP}_k \times \text{EM}_k) + \text{WTI} \times \text{PTI} + \text{WTD} \times \text{PTD} - \sum_{k=1}^{3} (p_1 \times x_{1k}) \right), (X_2 \times \text{ERW}_2 + \text{WTI} \times \text{ERW}_2 - \text{PTI} \times \text{WTI} - p_2 \times X_2)$ and $(X_3 \times \text{ERW}_3 + \text{WTD} \times \text{ERW}_3 - \text{PTI} \times \text{WTD} - p_3 \times X_3)$ represents the economic returns in different sectors. $\text{ERW}_k$ presents the economic return per unit of water consumption in crops, $\text{ERP}_k$ is the economic return from agricultural product exports, where $\text{ERP}_k = \text{ERW}_k \times \text{VW}_k$. In addition, $\sum_{k=1}^{3} (\text{ERP}_k \times \text{EM}_k), \sum_{k=1}^{3} (c_k \times \text{IM}_k)$ and $\text{TC} \times (\text{WTI} + \text{WTD})$ are the export return, import cost and transaction cost, respectively, where $c_k$ is the economic cost due to agricultural product imports, and TC is the transaction cost per unit of water resource from the agricultural sector to the industrial or domestic sectors.

Lower-level decision makers are water usage sectors that independently minimize their own vulnerabilities. In each water usage sector, water vulnerability is considered with respect to the water supplies and demands. In this way, the allocation strategy is expected to have the ability to meet the "water demands" and provide effective water withdrawal guidance to the water usage sectors. In this paper, the vulnerability, denoted as $F, F \geq 0$, is assessed from two aspects: destroying degree





$(F^{\mathrm{DD}})$ and economic loss $(F^{\mathrm{EL}})$, as shown in Eqs. (D4-D15). An $F$ equal to $0$ means that the water resource system is stable in the planned period. When F has a value greater than $0$, this implies greater vulnerabilities and reflects poor management and allocation.

Agricultural sector:  $\min F_{1k}$ $\qquad$ (D4)

$$F_{1k} = \omega^{\mathrm{DD}} F_{1k}^{\mathrm{DD}} + \omega^{\mathrm{EL}} F_{1k}^{\mathrm{EL}}, k = 1, 2, 3 \tag{D5}$$

$$F_{1k}^{\mathrm{DD}} = \frac{\max\left((d_{1k} - x_{1k}), 0\right)}{d_{1k}}, \tag{D6}$$

$$F_{1k}^{\mathrm{EL}} = \max\left((x_{1k} - d_{1k}), 0\right) p_1 \tag{D7}$$

Industrial sector:  $\min F_2$ $\qquad$ (D8)

$$F_2 = \omega^{\mathrm{DD}} F_2^{\mathrm{DD}} + \omega^{\mathrm{EL}} F_2^{\mathrm{EL}} \tag{D9}$$

$$F_2^{\mathrm{DD}} = \frac{\max\left(d_2 - (X_2 + \mathrm{WTI}), 0\right)}{d_2}, \tag{D10}$$

$$F_2^{\mathrm{EL}} = \max\left(((X_2 + \mathrm{WTI}) - d_2), 0\right) p_2 \tag{D11}$$

Domestic sector:  $\min F_3$ $\qquad$ (D12)

$$F_3 = \omega^{\mathrm{DD}} F_3^{\mathrm{DD}} + \omega^{\mathrm{EL}} F_3^{\mathrm{EL}} \tag{D13}$$

$$F_3^{\mathrm{DD}} = \frac{\max\left(d_3 - (X_3 + \mathrm{WTD}), 0\right)}{d_3}, \tag{D14}$$

$$F_3^{\mathrm{EL}} = \max\left(((X_3 + \mathrm{WTD}) - d_3), 0\right) p_3 \tag{D15}$$

where $\omega^{\mathrm{DD}}$ and $\omega^{\mathrm{EL}}$ are the weights for the two aspects in different sectors, for $\forall k = 1, 2, 3$.

## D2   Model Constraints

Specific constraints reflect the management rules and behaviors in real-world practice. Objective function (D1) is subject to constraints (D16)-(D19), while constraint (D21) characterizes the feasible region on the lower level.

### D2.1   Available Water Constraint

The water withdrawal for the three sectors cannot exceed the initial water gained by the irrigation district.

$$\sum_{i=1}^{4} X_i \leq \mathrm{AW} \tag{D16}$$

### D2.2   Price Constraint

When deciding the inner transactions between the sectors, the benefits also need to be considered. When the water demand in
the industrial or domestic sectors is greater than the water withdrawal, the water usage sector must buy water from the water market; conversely, the manager can sell extra water in the water market if water trading leads to greater benefits than achieved





by using that water to irrigate crops. Nevertheless, the transaction price should not exceed the water withdrawal price.

$$p_1 < \text{PTI} < p_2, \quad p_1 < \text{PTD} < p_3 \tag{D17}$$

### D2.3 Ecological Water Requirements

To guarantee the sustainable development of the whole river basin, the minimum ecological water requirements should be satisfied in the whole river basin.

$$X_4 \geq e \tag{D18}$$

### D2.4 Export and Import Balance Equation

The annual export volume plus the grain consumption should be smaller than the total grain yield plus the annual import
volume. Eq. (D19) gives the export-import balance equation. POP$\varpi_k$ is the total consumption in this irrigation area, POP is the population in Hetao irrigation area, $\varpi_k$ presents the annual crop $k$ consumption per unit in the Hetao irrigation area, and $l_k$ is the crop yield, which is decided based on the water allocated to the different crops on the lower level.

$$\text{EM}_k + \text{POP}\varpi_k \leq l_k + \text{IM}_k \tag{D19}$$

$$l_k = y_k \frac{x_{1k}}{W_k} \tag{D20}$$

### D2.5 Planting Area Constraint

The land area allocated to the different crops in a given cropping season must not exceed the total cultivable area (denoted by $A$) in that season, as shown in constraint (D21), in which $A_k = \frac{X_k}{W_k}$.

$$\sum_{k=1}^{3} A_k \leq A \tag{D21}$$

*Author contributions.* All of the authors helped to conceive and design the analysis. Zhongwen Xu and Xudong Chen developed the model
and wrote the paper. Liming Yao and Huijuan Wu contributed to the data collection and the writing of the paper.

*Competing interests.* The authors declare that they have no competing financial interests.

*Acknowledgements.* The main data sources for the solution are based on the Bayna Noaoer yearbook, Hetao irrigation district statistical data,
BayanNur Water Resources Bulletin and some published papers. Table B4 provides a general overview of the input variables, data sources
and determination in this study. Some are cited from the website and published papers directly, and others are predicted by relative equations
based on historical data.



We thank those that have given constructive comments and feedback to help improve this paper. Support was provided by the National Natural Science Foundation of China [Grant No. 71771157].



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
