# Peer review of "A novel data-driven analytical framework on hierarchical water allocation integrated with blue and virtual water transfers"

_Hydrology and Earth System Sciences, 2019_

## Referee Comment (RC1) · Anonymous Referee #1 · 2 Oct 2019

This paper developed a data-driven analytical framework for optimal water resource allocation that addresses blue virtual water transfers in the presence of different hydrological and economic conditions. Water resource utilization in district irrigation and conflicts between competing users and social-economic development and environment protection has substantially increased importance in recent years and the ideas proposed in this paper are interesting and valuable.

From my point of view, the work is well-done and provides interesting results to the HESS and thus it merits to be published. Just, I suggest some minor modifications before publication: 1. The conception of "blue water" and "virtual water" should be

clearly described. Do they have an inclusion relationship? 2. The proposed model focused on the irrigation district problem. So the title of this research article should be more specific to highlight the water allocation in the irrigation sector. 3. There are many useful conclusions drawn from the results of the proposed optimization. Maybe you should give some advice to the decision-makers facing similar problems.

---

## Short Comment (SC1) · 4 Oct 2019

The paper presents an interesting and innovative approach: the combination of game theory models to address water sustainability, what is applied to a case in China. For this reason the paper could be accepted for publication with the following contributions: (1) Two types of conflicts among one leader and multiple followers in the water resource management system are analyzed, which helps optimize the water allocation, withdrawal and transaction processes more comprehensively. (2) A novel game-theory model based on the Stackelberg game and Nash-Harsanyi equilibrium is developed for resolving the "leader-followers" and "competing followers" conflicts by strategic interaction. (3) Having incorporated the concepts of blue and virtual water transfers, our model is able to further relieve the water scarcity stress and offers insights on crop planting and import/export quantities.

---

## Author Comment (AC1) · 19 Oct 2019

1. The conception of "blue water" and "virtual water" should be clearly described. Do they have an inclusion relationship?

Response: Thank you for your question. Blue water is the surface or groundwater that runs off to the ocean, which is used for industrial and domestic purposes and irrigation in agriculture; while virtual water is the water embedded in a product (Allan et al., 1993). Virtual water content is the amount of water per unit of product that is consumed during the production process.

In addition, it's known that freshwater essentially stems from precipitation, which partitions into green and blue water (Falkenmark, 2013a). Virtual water content can be specifically divided into blue water content and green water content. In this paper, blue water transfer is regarded as a means of reallocating water among sectors, it can be directly transferred from one sector to another through conveyance infrastructure after each sector (e.g., domestic, industrial or agricultural) has been granted temporary water withdrawal rights. Virtual water transfer is characterized by crops trade. It helps import countries save the volume of water necessary to produce a certain commodity.

Reference

[1] Allan J A. Fortunately there are substitutes for water otherwise our hydro-political futures would be impossible. Priorities for water resources allocation and management, 1993, 13(4): 26.

[2] Falkenmark M. Growing water scarcity in agriculture: future challenge to global water security. Philosophical Transactions of the Royal Society A: Mathematical, Physical and Engineering Sciences, 2013, 371(2002): 20120410.

2. The proposed model focused on the irrigation district problem. So the title of this research article should be more specific to highlight the water allocation in the irrigation sector.

Response: Thank you for your comment, we changed the title from the original one to "A novel data-driven analytical framework on hierarchical water allocation integrated with blue and virtual water transfers: A case study of China" after careful consideration.

3. There are many useful conclusions drawn from the results of the proposed optimization. Maybe you should give some advice to the decision-makers facing similar problems.

Response: Thank you for your suggestion. It's important to give managerial insights to decision makers who may encounter the same problem.

HESSD

Interactive
comment

In many places, including the north of China, there are two different hierarchical structures, the water affairs bureau and the water usage sectors, within an irrigation area. Against the backdrop of water scarcity, incommensurable conflicts exist among different water users and the water affairs bureau because of differing objectives. Additionally, faced with multiple followers, another main problem is that various water usage sectors, such as agricultural, industrial, domestic and ecological sectors, compete for limited water resources. Due to the unsuitability of this problem for modeling by conventional methods, a novel game model is presented considering the water allocation and blue/virtual transfers together, in view of the hierarchical structure of the problem.

In terms of management insight, some policy implications have been provided in the manuscript based on the results solved by the proposed model. In the response letter, we listed some of them.

1) blue water transfer is suggested in areas with uneven water distribution condition, which provides an opportunity for each sector to achieve an efficient utilization of water. To be specific, some initial water rights could be transferred from the agricultural sector to the industrial and domestic sector.

2) virtual water transfer is suggested in (semi)-arid areas, which provides a new opportunity for water conservation and land saving. To be specific, key crops, particularly water-intensive crops (e.g., wheat and sunflower), could be imported from other countries rather than being grown domestically.

---

## Editor Comment (EC1) · Pieter van der Zaag (Editor) · 26 Oct 2019

The short comment submitted by Xudong Chen (email: chenxudong198401@163.com) published on 4 October 2019 is inappropriate, as authors are not supposed to make "Short Comments". Authors should use the "Author Comment" option if they want to contribute a comment. It is disturbing that in this short comment Xudong Chen did not disclose co-authorship of the paper this comment addresses.

Pieter van der Zaag, the editor handling this submission

---

## Short Comment (SC2) · 31 Oct 2019

A novel data-driven analytical framework on hierarchical water allocation integrated with blue and virtual water transfers

The authors proposed a bi-level methodology that combines two game theoretical models: Stakelberg competition and Nash-Harsanyi bargaining to optimize the water usage. The model is applied to a case study in the Chinese region of Hetao, a sensitivity analysis is carried and policy making insights are obtained. In my opinion the idea of combining Stakelberg competition and Nash-Harsanyi bargaining models to deal with the problem of water management is interesting and original, as far as this referee knows,

for this reason the paper could be accepted for publication. Therefore minor revision of the current version of the paper is needed. Comments and suggestions are given below.

1. I do not understand why the authors need the disagreement point of the leader in the Stakelberg model because the leader does not take part in the bargaining procedure of the lower level. Please, add some more explanations. 2. Conclusion section is a little lengthy and therefore needs to be shrunk. Too much information would distract people's attention. 3. The heading titles should be revised, such as Section 5.1 Which sector largely contributes to water stress.

---

## Referee Comment (RC2) · Muhammad Hashim (Referee) · 18 Feb 2020

The research entitled "A novel data-driven analytical framework on hierarchical water allocation integrated with blue and virtual water transfers" is interesting and provides some insights that can be used by the decision makers for taking sustainable decisions. However, The paper could be accepted for publication if the authors incorporate the following comments and suggestions in final draft . 1. Elaborate the disagreement point of the leader in the Stakelberg model. 2. I strongly recommend to explain the meaning of all variables and parameters for improving the readability of the paper. 3. What are the termination conditions mentioned at Page 11? Please make it clear. 4.

Line 273, please revise the formation of reference citation. 5. Variable unit is needed in each Table, such as Table 8. 6. Unnecessary reference lump should be avoid through the whole paper, such as "Generally speaking", cited references should have meaning. 7. Conclusion section is a little lengthy and therefore needs to be shrunk. Too much information would distract people attention. 8. Personally, I would like to see some more elaboration on managerial insights. What kind of problem can be solved by the proposed model? Whether the results are able to offer some suggestions on future water management in real-world practice.

---

## Author Comment (AC2) · 24 Feb 2020

The research entitled "A novel data-driven analytical framework on hierarchical water allocation integrated with blue and virtual water transfers" is interesting and provides some insights that can be used by the decision makers for taking sustainable decisions. However, the paper could be accepted for publication if the authors incorporate the following comments and suggestions in final draft.

[Comment 1.] Elaborate the disagreement point of the leader in the Stakelberg model.

[Response.] Thank you for your comment. We have provided additional information

regarding the disagreement points for better clarity in the Solution Procedure section.

First, we introduced the importance of defining a disagreement point. Combined with the bi-level Stakelberg model, there was an interactive process between the leader and followers. The leader possesses a higher priority to move first, and the followers play among themselves according to the Nash-Harsanyi equilibrium after observing the leader's announced strategy; then, the followers provide feedback to the leader. The leader then maximizes its objective function based on the identified best-response strategies of the followers. During the bilevel strategic interaction, with the decrease in sectoral vulnerability at the lower level, the water utilization efficiency of the system at the upper level will decrease to some extent. Once the solution is worse than the disagreement point, irrespective of whether it is for the leader or the competing followers, the decision maker can no longer accept it.

Second, we introduced the derivation of the disagreement point and the application in this paper. Gao & Lv, (1989) introduced the interactive satisficing trade-off method based on the tactics of the ideal point method, which helped to resolve problems including multiple conflicting objective functions. The ideal point was the situation in which the objective function reached its optimal value; however, in general, not all objective functions would reach the ideal points simultaneously. A set of ideal points was not in the feasible set, but rather each objective value would exist between the negative ideal point and positive ideal point. Previously, Hans and Eric, (1991) described how the disagreement point approach can be applied to bargaining solutions. Furusawa and Wen, (2002) analyzed the tariff trade war and pointed out that the disagreement point was regarded as bargaining frontier in the bargaining process.

As for the biobjective models (Gao & Lv, 1989), a solution that was simultaneously optimal for each decision maker's objective function rarely occurred among the bilevel problems involving multiple decision makers, because of conflicts among the leader and the competing followers. Hence, by utilizing the concept of the bargaining game and negative ideal point in this paper, we extended the definition of the disagreement

point, namely, the disagreement point that presented the worst result, which the decision maker was unwilling to accept. Hence, additional constraints for each level were added for which each objective function value was better than the respective disagreement point.

Finally, combined with the practical problem described in this paper, we have defined the vector of the disagreement points as the maximum vulnerability to the followers and the minimum efficiency to the leader. To be specific, the disagreement point of each objective was calculated at page 11. References Gao, J., & Lv, X. Interactive satisfying trade-off method for multiobjective optimization (in Chinese). Journal of Hefei University of Technology, 1989, (2), 32-41. Hans P, Eric V D. Characterizing the Nash and Raiffa Bargaining Solutions by Disagreement Point Axioms. Mathematics of Operations Research, 1991, 16(3):447-461. Furusawa T, Wen Q. Disagreement points in trade negotiations. Journal of International Economics, 2002, 57(1):133-150.

[Comment 2.] I strongly recommend to explain the meaning of all variables and parameters for improving the readability of the paper.

[Response.] Thank you for the detailed comment. We have enriched Appendix A to explain the meaning of all variables and parameters in details (Please see the supplement file AppendixA.pdf). In addition, appendix B and C are given to elaborate the details of model proposal.

[Comment 3.] What are the termination conditions mentioned at Page 11? Please make it clear.

[Response.] The termination condition in each case is that when individuals stop evolving, that is, when the best solution (individual) of the current generation is the same as the previous generation or when the algorithm reaches one hundred iterations (Messias et al., 2016). Reference Messias, Valter Rogério, Estrella J C , Ehlers R , et al. Combining time series prediction models using genetic algorithm to autoscaling Web applications hosted in the cloud infrastructure. Neural Computing and Applications,

2016, 27(8):2383-2406.

[Comment 4.] Line 273, please revise the formation of reference citation.

[Response.] Thank you for the detailed comment, we have corrected the mistake and check the whole manuscript.

[Comment 5.] Variable unit is needed in each Table, such as Table 8.

[Response.] Thank you for the detailed comment, we have added the unit for each table.

[Comment 6.] Unnecessary reference lump should be avoid through the whole paper, such as "Generally speaking", cited references should have meaning.

[Response.] Thanks for your suggestion. We have read the whole paper again and enrich the descriptions for cited references and deleted some unnecessary references.

[Comment 7.] Conclusion section is a little lengthy and therefore needs to be shrunk. Too much information would distract people attention.

[Response.] Thank you for your suggestion. We have condensed the conclusion section in the revised manuscript following the layout including problem description, model, application results, comparative and sensitivity results, managerial insights and future research directions.

Conclusion

In this study, we proposed a novel model based on Stackelberg-Nash-Harsanyi game theory for analyzing the water reallocation problem in the promise of water transfers and crops transactions. To describe realistic water allocation, withdrawal and transaction processes, this paper employed a bilevel framework with one leader and multiple followers, where a water affairs bureau was in the leadership position and multiple water usage sectors were in a lower position. Vulnerability, including the destroying degree (caused by deficient water withdrawal) and economic loss (caused by excess
water withdrawal), is the lower level objective. In addition, the water utilization efficiency was maximized at the upper level across the whole system.

Blue and virtual water transfers are examples of mechanisms that essentially relieve uneven distribution of water. In this study, one incorporated idea is that the blue water can be reallocated to industrial and domestic sectors to develop water-saving agriculture. The other incorporated idea is that the virtual water existing in crops is quantified to optimize inter-regional exports and international imports. Virtual water transfer path is from regions defined by comparative advantage of water resources to those defined by comparative disadvantage of water resources so that water utilization efficiency was improved for the regions involved. Besides, blue water transfer path is from sectors with lower water vulnerability value to those with higher water vulnerability value so that vulnerability caused by exceed/lack the quantity of water demand decreased for the sectors involved. All told, having incorporated the concepts of blue and virtual water transfers, our model is able to further relieve the water scarcity stress as well as reduce the vulnerability and increase the water utilization efficiency. During the dynamic strategic interaction, the leader modifies its decision based on the best-response strategies by the followers.

To verify the feasibility and practicality of the developed model, a real-world application was conducted in the Hetao irrigation district. The results found that water demand in the domestic sector was first satisfied, followed by that of the agricultural and industrial sectors; blue water transfer provided an opportunity for each sector to achieve an efficient utilization of water, and virtual water transfer provided a new opportunity for water conservation and land saving. To be specific, some initial water rights were transferred from the agricultural sector to the industrial and domestic sectors, and key crops, particularly water-intensive crops (e.g., wheat and sunflower), were imported from other countries rather than being grown domestically.

Furthermore, to demonstrate the superiority of the developed model, two comparative scenarios were considered: one without virtual water transfers and the other limiting

blue water transfers. The analysis of Scenario A highlighted the importance of virtual water transfers in significantly alleviating water usage stress. Furthermore, the analysis of Scenario B verified that blue water transfers reduced vulnerability in each sector and demonstrated how inter-regional exports, international imports and blue water transfers can result in higher water utilization efficiencies in all three sectors.

Considering the changing environment and water-saving strategies, several scenarios were assessed, including one without any change (the results of the proposed model). Faced with varying water availability, it was concluded that in wet environments, crops with more virtual water could be planted and exported. In contrast, in dry environments, crops that require more irrigation should be imported.

An analysis of how to reduce water demand in different water usage sectors is discussed. The results found that blindly decreasing domestic water demands can harm efficiency and increase vulnerability even though this kind of strategy can reduce total consumption in the water allocation system. When industrial water demand decreased by 10\%, the overall efficiency showed improvement; however, too great of a reduction could result in a negative influence on the sustainability of the water allocation system. With a decrease in agricultural water demand, water utilization efficiency improved, and total consumption decreased. However, if there is a very large reduction, the industrial sector vulnerability would be aggravated.

Hence, several policies can be implemented in terms of various sectors to alleviate the regional water stress and the negative impact caused by the crop trade, as well as to safeguard regional food security to develop a sustainable irrigation district.

Further, under the changing market in terms to crop transaction price and water transfer price, several scenarios were conducted to probe deeper the optimal allocation and transfers strategies. The results denoted that the model was more sensitive to available water and sectoral demand, instead of crop import price. Further, the model was more sensitive to water price variation in maize, followed by wheat and sunflowers. Then

the fitted smoothing lines based on the solved values implied that the price elasticity of water transaction is not linear, which gave a direction for formulation demand function in future research.

This work focused on water allocation in a bilevel framework. Given that we explored only one leader and multiple followers, there are possibilities for multiple leaders and multiple followers in future research. Furthermore, a dynamic analysis from the time perspective may be interesting.

[Comment 8.] Personally, I would like to see some more elaboration on managerial insights. What kind of problem can be solved by the proposed model? Whether the results are able to offer some suggestions on future water management in real-world practice.

[Response.] Thank you for your question. We give the following explanation.

In many places, including the north of China, there are two different hierarchical structures, the water affairs bureau and the water usage sectors, within an irrigation area. Against the backdrop of water scarcity, incommensurable conflicts exist among different water users and the water affairs bureau because of differing objectives. Additionally, faced with multiple followers, another main problem is that various water usage sectors, such as agricultural, industrial, domestic and ecological sectors, compete for limited water resources. At present, there is very little literature on irrigation districts that take into account blue/virtual water transfers and water allocation simultaneously within a bi-level framework. Due to the unsuitability of this problem for modeling by conventional methods, a novel game model is presented considering the water allocation and blue/virtual transfers together, in view of the hierarchical structure of the problem.

Based on the conceptual framework, this paper has proposed a novel game model based on Stackelberg game and Nash-Harsanyi equilibrium models to analyze the one-leader multi-follower problem, including the water affairs bureau and sectoral water users in water resource management systems. In this way, the water affairs bureau

possesses a higher priority to decide the initial water allocation rights, and the followers play among themselves according to the Nash-Harsanyi equilibrium after observing the leader's announced strategy and provide feedback to the leader. The leader then maximizes the system's water utilization efficiency based on the identified best-response strategies of the followers. The process proceeds with the interactive Stackelberg game as well as the Nash-Harsanyi bargaining model. Finally, through a strategic interaction, an optimal solution can be gained.

Being a water-scarce area, the Hetao irrigation district is still a grain exportation area with a large amount of output. Since the reform and opening-up, China's crop production center began to shift from the originally developed southern and eastern regions to the economically backward northern and western regions. Additionally, crops imported from other countries, accompanied by virtual water transfers, have become another measure to relieve the local water scarcity pressure. However, in previous articles, scholars have ignored that a great amount of crop imports would lead to a decrease in local production, which could significantly reduce the income of local farms. Hence, this paper endeavors to formulate a strategic plan by employing virtual water to reduce crop water demands jointly with blue/ green water irrigation systems.

By incorporating virtual/ blue water transfers into the above proposed model, three realistic problems can be solved: (1) optimizing the water withdrawals in agricultural, domestic and industrial sectors and determining the irrigation requirements for different crops in a planning year; (2) optimizing the virtual water (existing in crops) quantities to be imported and exported, and (3) optimizing the blue water quantities to be transferred from the agricultural to non-agricultural sectors.

We applied the developed model to a real-world practice to acquire more information on water saving, land planning and sustainable development. In addition, It can be applied to any water resource system not only in the Hetao irrigation district but also in other areas with similar problems.

The managerial insights derived from results of the case study are showed as follows:

(1) crop consumption in this district is far less than that produced, particularly for sunflowers. The surplus crops could be sold to south China.

(2) Pressure on regional water resources was relieved, regional food security was safeguarded and planting land was appropriately utilized. According to the results, the total land use for crops was reduced by 16.7% due to the international imports; further, the total virtual water in the imported crops was calculated as 4.16×109 m3, which would save 75.64% of the total available water in the area.

(3) The negative impact caused by the crop trade was alleviated, and the potential water saving strategy was fully explored. According to the results, after considering agricultural blue water transfers and virtual water transfers, an increase in the water resource system efficiency and a decrease in vulnerability values were found in each sector, as shown in Table 4.

Hence, the above analysis suggested that when there was insufficient water, blue water could be transferred to the industrial and domestic sectors from the agricultural sector to enhance the water utilization efficiency and achieve greater economic benefits.

(4) Furthermore, to provide more information to decision makers, we conducted the following work.

First, the domestic sector with the value of 0.254 seemed to be the direct reason for water stress in this district. Before drawing up a long-term future water plan, the determination of which sector was the largest contributor to regional water stress is of great importance.

Second, virtual/ blue water transfers were verified to be beneficial for both hierarchical decision makers, for which the results are shown in Figs. 8-10 and Table 6. Compared with Scenario A limiting virtual water transfers, the total water consumption of the proposed model decreased from 4,547,691,062 m3 to 4,446,716,331 m3; the blue water

transfer value was reduced from 1,126,120,905 m3 to 1,085,970,257 m3. Compared with Scenario B, the total water consumption solved by the proposed model increased from 4,327,537,703 m3 to 4,446,716,331 m3, with the most water being consumed by the domestic sector rather than the agricultural sector, which implied that there was a positive effect propelling the water utilization efficiency and reducing the sector vulnerability when blue water transfers were considered.

Third, scenario analysis was conducted to cope with the water management strategy under climate and hydrological changes. Table 8 shows the solutions to the decision variables under the varying available water constraints. The analysis also demonstrated that industrial vulnerability is the least sensitive to changeable climate change, followed by the agricultural sector, with the domestic sector being the most sensitive. Overall, the domestic sector should be paid more attention regardless of the presence of extreme dry or wet conditions, as the sector is more sensitive to the available water.

Furthermore, with a decrease in available water, the import trade increases, and with an increase in available water, there is an increase in the export trade, particularly in sunflower exports (a crop that provides higher economic benefits with greater water consumption). Hence, in a wet environment, crops that contain greater virtual water are suggested to be planted and exported; in a dry environment, it is suggested that crops that require more irrigation be obtained through import trading.

Fourth, to regulate water saving precisely, several sensitivity analyses were conducted to address the purpose of water saving in different sectors, which provided insights into future investments into water conservation and sustainability development. In addition, by using an Extended Fourier Amplitude Sensitivity Test (EFAST) method, we found that the proposed model was less sensitive to the import crop prices than to the available water and sectoral water demand.

Overall, under the bi-level framework, individual and system benefits can be balanced. In addition, measures solved by the proposed mode have a greater capacity to

alleviate the pressure on regional water resources, safeguard regional food security, alleviate the negative impact caused by the crop trade and develop a sustainable irrigation district.

Please also note the supplement to this comment:
https://www.hydrol-earth-syst-sci-discuss.net/hess-2019-389/hess-2019-389-AC2-supplement.pdf

**Supplement:**

**Appendix A: Notations**

The following notations are used to develop the model.

**Indexes:**

$k$ : crop indicators, $k = 1$ for wheat, $k = 2$ for maize, $k = 3$ for sunflower

$i$ : water usage indicator, $i = 1$ for agricultural sector, $i = 2$ for industrial sector, $i = 3$ for domestic sector, $i = 4$ for ecological sectors

**Parameters:**

$p_i$ : water price set by the leader for sectoral water uses, RMB/m$^3$

$\text{ERW}_i$ : economic return per unit of water consumption in sectors, $i = 1,2,3$, RMB/m$^3$

$\text{ERP}_k$ : economic return from agricultural products exports, $i = 1$, where $\text{ERP}_k = \text{ERW}_k \times VW_k$, RMB/m$^3$

$c_k$ : economic costs because of agricultural products imports RMB/kg

TC : transaction cost per unit of water resource from agricultural sector to industrial or domestic sectors RMB/m$^3$

$\mu$ : the irrigation coefficient, which presents the utilization effectiveness of irrigation water

$A$ : total available area for crop planting, hm$^2$

$\phi_{\text{ind}}$ : the gross industrial output value, RMB

$R_k$ : effective rainfall, mm

$\varpi_k$ : crop $k$ consumption per unit in the Hetao irrigation area, kg/person

**Auxiliary variables (continuous variables)**

AW : maximum volume of available water in Hetao irrigation district, m$^3$

PTI : price of water transfers to industrial sector, RMB/m$^3$

PTD : price of water transfers to domestic sector, RMB/m$^3$

$w_k$ : water irrigation for crop $k$, /m$^3$

$W_k$ represents the blue and green water components in crop $k$, m$^3$

$y_k$ represents the crop yield per unit of irrigation area, kg/hm$^2$

$l_k$ : total yield of crop $k$, $kg$

$\text{VW}_k$ : virtual water content of crop $k$, m$^3$ kg$^{-1}$

$A_{1k}$ : area allocated to crop $k$, hm$^2$

$d_i$ : water demand of sectors, $i = 1,2,3,4$, m$^3$

$d_{1k}$ : water demand of crops in agricultural sector, $k = 1,2,3$, m$^3$

$\phi_{\text{pop}}$ : per capita disposable income, RMB

POP: population in the Hetao irrigation area

**Decision variables (continuous variables)**

$X_i$ : initial water rights in sectors, $i = 1,2,3,4$, determined by the upper-level decision maker, m$^3$

$\text{EM}_k$: quantity of products exports, determined by the upper-level decision maker, kg

$\text{IM}_k$: quantity of products imports from international trade, determined by the upper-level decision maker, kg

WTI : water transfer from agricultural sector to industrial sector, determined by the lower-level decision makers, $\text{m}^3$

WTD : water transfer from agricultural sector to domestic sector, determined by the lower-level decision makers, $\text{m}^3$

$x_{1k}$ : water irrigated to crop $k$ in the agricultural sector, determined by the lower-level decision maker, $\sum\limits_{k=1}^{3} x_{1k} = X_1$, $\text{m}^3$

---

## Author Comment (AC3) · 24 Feb 2020

The authors proposed a bi-level methodology that combines two game theoretical models: Stakelberg competition and Nash-Harsanyi bargaining to optimize the water usage. The model is applied to a case study in the Chinese region of Hetao, a sensitivity analysis is carried and policy making insights are obtained. In my opinion the idea of combining Stakelberg competition and Nash-Harsanyi bargaining models to deal with the problem of water management is interesting and original, as far as this referee knows, for this reason the paper could be accepted for publication. Therefore minor revision of the current version of the paper is needed. Comments and suggestions are given

below.

[Comment 1.] 1. I do not understand why the authors need the disagreement point of the leader in the Stakelberg model because the leader does not take part in the bargaining procedure of the lower level. Please, add some more explanations.

[Response.] Thank you for your comment. We have provided additional information regarding the disagreement points for better clarity in the Solution Procedure section.

First, we introduced the importance of defining a disagreement point. Combined with the bi-level Stakelberg model, there was an interactive process between the leader and followers. The leader possesses a higher priority to move first, and the followers play among themselves according to the Nash-Harsanyi equilibrium after observing the leader's announced strategy; then, the followers provide feedback to the leader. The leader then maximizes its objective function based on the identified best-response strategies of the followers. During the bilevel strategic interaction, with the decrease in sectoral vulnerability at the lower level, the water utilization efficiency of the system at the upper level will decrease to some extent. Once the solution is worse than the disagreement point, irrespective of whether it is for the leader or the competing followers, the decision maker can no longer accept it.

Second, we introduced the derivation of the disagreement point and the application in this paper. Gao & Lv, (1989) introduced the interactive satisficing trade-off method based on the tactics of the ideal point method, which helped to resolve problems including multiple conflicting objective functions. The ideal point was the situation in which the objective function reached its optimal value; however, in general, not all objective functions would reach the ideal points simultaneously. A set of ideal points was not in the feasible set, but rather each objective value would exist between the negative ideal point and positive ideal point. Previously, Hans and Eric, (1991) described how the disagreement point approach can be applied to bargaining solutions. Furusawa and Wen, (2002) analyzed the tariff trade war and pointed out that the disagreement point

was regarded as bargaining frontier in the bargaining process.

As for the biobjective models (Gao & Lv, 1989), a solution that was simultaneously optimal for each decision maker's objective function rarely occurred among the bilevel problems involving multiple decision makers, because of conflicts among the leader and the competing followers. Hence, by utilizing the concept of the bargaining game and negative ideal point in this paper, we extended the definition of the disagreement point, namely, the disagreement point that presented the worst result, which the decision maker was unwilling to accept. Hence, additional constraints for each level were added for which each objective function value was better than the respective disagreement point.

Finally, combined with the practical problem described in this paper, we have defined the vector of the disagreement points as the maximum vulnerability to the followers and the minimum efficiency to the leader. To be specific, the disagreement point of each objective was calculated at page 11.

References

Gao, J., & Lv, X. Interactive satisficing trade-off method for multiobjective optimization (in Chinese). Journal of Hefei University of Technology, 1989, (2), 32-41. Hans P, Eric V D. Characterizing the Nash and Raiffa Bargaining Solutions by Disagreement Point Axioms. Mathematics of Operations Research, 1991, 16(3):447-461. Furusawa T, Wen Q. Disagreement points in trade negotiations. Journal of International Economics, 2002, 57(1):133-150.

[Comment 2.] 2. Conclusion section is a little lengthy and therefore needs to be shrunk. Too much information would distract people's attention.

[Response.] Thank you for your suggestion. We have condensed the conclusion section in the revised manuscript following the layout including problem description, model, application results, comparative and sensitivity results, managerial insights and future

research directions.

[Comment 3.] The heading titles should be revised, such as Section 5.1 Which sector largely contributes to water stress.

[Response.] Thank you for your suggestion. We have revised the unsuitable section name. For example, heading title of section 5.1 was changed to "Main reasons for water stress".

---

## Author Response (AR1)

Response to comments

*General Comment:* ***The manuscript has been reviewed by two reviewers, and the authors have, in my view, responded adequately to the issues raised. I have, however, additional comments that were not raised by the reviewers, and which I like to share with the authors, and I invite them to react to these. My comments are quite significant. When the authors submit a revised manuscript I may decide to send it again out for review. The paper addresses an important and interesting issue, namely how to deal with water scarcity taking both blue water transfers and virtual water transfers into account, in a 2-level decision setting, and also taking account of three different water using sectors. Quite a complex setting, and an ambitious undertaking. Given such complexity it is important that the argument is clearly presented and here the manuscript needs to be improved significantly (towards the end of this comment I give details what in my view should be addressed to improve the readability of the paper).***

Response: We would like to thank you and the two reviewers for the valuable comments and suggestions on our manuscript, all of which have assisted us in substantially improving the paper, as detailed in the following responses.

*Comment 1-1:* ***A first comment is that the paper makes no attempt whatsoever to validate the proposed model, or at least to show that model outcomes are plausible; this could have been done, for example by comparing model results with observed data, and discussing similarities and differences.***

Response: Thank you for your observation. As described in the manuscript, the basic idea was that the two-decision maker levels compromise to determine an optimal global solution as it is impossible that both can simultaneously achieve their desired goals. The proposed model was solved using the MATLAB R2017b solver, with the iteration process given in Figure 1. To be specific, the leader makes the first decision, and based on the leader's solution, each follower seeks an optimal solution within the necessary compromise. Then, the followers send their solutions to the leader, who then adjusts their own goals and preferences. The chosen upper level optional solution standard was that the value of the objective function needed to be larger than previous value and the chosen lower level optional solution standard was that one of the objective function values was smaller than the previous objective function value. This iterative process continues until the termination condition is satisfied, which is when the average change in the fitness value

is less than the options or when the algorithm reaches one hundred iterations.

[Figure]

Figure 1 Iteration process for the upper level objective function.

In the revised manuscript, the outcomes were compared with data from a planning report. In **Section 4.3 Optimal Virtual Water Trade**, Figure 5 compares the results from the proposed model with those reported in the Water Resources Planning Report for Bayan Nur, from which it was found that as the Hetao irrigation district was water-scarce, it was most suitable for growing maize, which accounted 182,424 hm$^2$, with a further 40% devoted to irrigated sunflowers, and 15% to irrigated wheat. In the Water Resources Planning Report of Bayan Nur, a 312,380 hm$^2$ area was planned for sunflowers, and 124,913 hm$^2$, 34,073 hm$^2$ for maize and wheat. In general, the main land use differences were related to the agricultural water usage sector. Economically, the total net import costs were 5.66 $\times 10^9$ CNY, with the saved water transferred to industrial or domestic sectors totaling about 3.35 $\times 10^{11}$ CNY. Therefore, from both water conservation and economic development perspectives, virtual water imports need to be included because of inter-regional exports and local consumption.

[Figure]

Figure 5. Predicted crop areas in the planning report compared to the optimal values (hm²)

*Comment 1-2: **Similarly, the authors could have conducted sensitivity analyses to verify how sensitive model results are to changes in input values of certain critical parameters; not for the authors to draw far reaching conclusions, but rather to re-assure the reader of the validity of the model. In the current manuscript the authors have indeed conducted several sensitivity analyses (on water availability, sectoral water demands, prices of import crops and water price), and they draw far reaching conclusions on the outcomes, without even trying to explain these outcomes (see below). But this is not convincing to me. It would be much more convincing to use sensitivity analysis to demonstrate the robustness and plausibility of the model.***

Response: Thank you for your suggestion. In the original manuscript, several sensitivity analyses were included and far reaching conclusions and managerial insights given. In the revised manuscript, based on your suggestion, we have rewritten this section to systematically investigate the allocation responses to changing values in the model's input and drastic changes in the model's structure. Two robustness measures: minimum water utilization efficiency and maximum sectoral vulnerability: were used to assess the optimal solution.

*Water availability.* The optimal results indicated that with a decrease in available water, the water utilization efficiencies fluctuated slightly, which verified the robustness of the model. When the amount of available water increased, the utilization efficiency fluctuated with an increase in total water consumption, and compared to the water utilization efficiency, the vulnerability values were within an acceptable range. Therefore, it was found that after the blue and virtual water transfers, the water allocation and import/export

strategies had the ability to adapt to varying quantities of available water,.

*Water demand.* With a decrease in the agricultural water demand, the water utilization efficiency improved and total consumption decreased. To monitor the performance and assess the responses to each scenario, Table \ref{tab:com4} indicated that the proposed model was robust to agricultural and industrial sector parameter changes, but was somewhat weaker when domestic water demand changed.

*Import prices.* When the imported crop import prices changed, there was only a small change in the water utilization efficiency and vulnerability, which also indicated that the proposed model was comparatively robust to changing import prices; however, it was found to be more sensitive to water price variations for sunflowers, followed by maize and wheat.

*Water prices.* Table 7 indicated that the optimized decision variables were less sensitive to the changes in market prices, and verified that the proposed model had a robustness to market prices.

More content describing the robustness of the proposed model was added in **sub-sections 5.4.1-5.4.4 (Sensitivity analysis)** in the revised manuscript.

*Comment 1-3*: ***The comparison of model results with scenarios omitting virtual water transfers and blue water transfers and with the two-staged model (section 5.2) is interesting, although it is not clear to me how the proposed model differs from the two-stage model. I would suggest to use this comparison to demonstrate the validity (and value) of the proposed model; rather than to formulate far-reaching conclusions. First the model must be validated before it is used to draw conclusions.***

Response: Thank you for your constructive suggestions. We have added models (C1) and (C2) to the Appendix in the revised paper to elaborate the differences in the models and have rewritten and condensed sections 5.2 and 5.3 to demonstrate the validity (and value) of the proposed model.

In general, the proposed model, and models (C1) and (C2) were all able to solve different realistic problems. Water-scarce areas such as the case study area in northern china have more land and have a duty to provide Southern China with crops. However, planting and crop irrigation strategies have been affected as importing water-intensive crops from other countries has become more common. Therefore, it is vital that water supply system models be developed that can determine the optimal quantities of allocated water and blue and virtual water transfers in hierarchical decision- making structures consisting of a water affairs

bureau and water usage sectors. The following paragraphs detail the model differences.

In model (C1), as the crop imports and exports are not considered, the decision variables are $X_i, \mathbf{WTI}, \mathbf{WTD}, x_{1k}$. Therefore, the crops are grown only for self-consumption rather than for the development of the agricultural economy. In model (C2), as the blue water transfers from the agricultural sector to the non-agricultural sectors are not considered, the decision variables include $X_i, \mathbf{EM}_k, \mathbf{IM}_k, x_{1k}$. Therefore, the water market is ignored, which hinders cooperative economic planning and market regulation development in China.

Given that water utilization efficiency, sectoral vulnerability and water stress are the key indexes for model robustness, it was found that the incorporation of the virtual water and blue water transfers into the traditional bilevel optimization model was able to alleviate water stress, improve water utilization efficiency, minimize sectoral vulnerability and increase system stability. Additional content can be found on **Page 19**.

The main difference between the two-stage optimization model and the proposed model was that a Stackelberg game between a leader and followers was fully considered in the proposed model and there were strategic interactions before the compromised solution was obtained, which were directly included in the solution paradigm and Pseudo code (steps 3 and 4).

[Figure]

Figure 4. Steps for solving the proposed model

**Table B1.** Pseudo code for the proposed model.

**Input:** The correlation parameter, disagreement points ($dis^U$, $dis_i^L$), bargaining weights ($\alpha_i$) ;
**Output:** A satisfactory solution and objective functions for the proposed model;
**Step 1:** Initialize population, construct fitness functions for the upper objective (Eff) and an auxiliary fitness function for the Nash-Harsanyi model in the lower level ($\prod_{i=1,2,3} \left( F_i - dis_i^L \right)^{\alpha_i}$);
**Step 2:** Randomly generate an initial solution, maximize the fitness function in the lower level
    IF (any lower level objective functions are lower than their disagreement points)
    {generation++;
    population [generation+1]=Select (Mutate(Crossover (parents)))};
    Else: go to step (3);
**Step 3:** Input the satisfactory solution from step (2), maximize the fitness function in the upper level
    IF (upper level objective function is lower than its disagreement point)
    {generation++;
    population [generation+1]=select (mutate(crossover (parents)))};
    Else: keep the current decision variables and corresponding solutions and go to step (4);
**Step 4:** Feedback
    IF(the updated lower level objective functions are still better than the disagreement points)
    {IF (generation > Maximum generation )
     {Output all the decision variables
     End loop}}
    Else: go to step (2)

*Comment 2-1: A second comment is that the entire section 5.4 on sensitivity analyses raises more questions than it answers. For example, consider the available water (Table 5): if I understand it well, when there is 10% more water available, less water is allocated to agriculture ($X_1$), and when there is 15% more water than the base case, more water is allocated to agriculture, and when there is even more water (20% more than the base case) the volume allocated to agriculture is suddenly halved. This begs for an explanation. Similar questions may be raised concerning the water allocated to the domestic sector ($X_3$).*

Response: Thank you for your detailed comment. We have enriched the explanation for the changes in water use in the discussion on model robustness. In the results, the amount of available water was the main factor regulating and controlling the allocation ratios between the different water users. From the values in the two robustness indexes, the proposed model may not be suitable for solving water allocation problems or blue and virtual water transfers as the imbalance in the market would attract the Chinese interventionist government's attention. In other hydrological environments, there are different water use strategies. For example, when there is 10% more water available through precipitation during a crop growth cycle, there would be less irrigation water needed, which means that less water would be allocated to agriculture. However, if there was 15% more water than the base case, more water would be allocated to agriculture because of an increase in crop exports. Because of Hetao irrigation district's comparative land advantage (Zhao et al., 2019), it is a suitable place to plant crops if there are enough water resources.

When there is 10% more water available, less water is allocated to agriculture and less agricultural water is sold to the non-agricultural sectors. When there is 15% more water than the base case, there is less water being allocated to the domestic sector as rainfall in a planning year has no positive influence on domestic use. In future research, it would be interesting to study the potential impact of an increase in available water on the domestic wastewater reuse rate and domestic water use efficiency (Ding et al., 2012; Li, 2014).

Response: Thank you for your suggestions. In the revised manuscript, some figures have been deleted and others have been revised, as follows. The crop proportions for each form: production, imports, exports and consumption: are shown on the left of Figure 6, and the virtual water content is shown on the right of Figure 6 to reflect the amount of embedded water in each crop in each form (production, import, export and consumption). Figure 7 (Figure 9 in the original manuscript) compares the water stress in different scenarios. Figure 7 (B)-(D) details the water stress in each sector and each crop in each of the different scenarios. In Figure 8 (D) (Figure 10 in the original manuscript), the virtual water embedded in the export crops is shown below the X-axis, and the virtual water embedded in the import crops is shown above the X-axis.

[Figure]

Figure 6 Inter-regional export and international import trade (m$^3$)

[Figure]

Figure 7. Water stress comparisons. (A) Total water stress; (B) Scenario B water stress; (C) Scenario A water stress; (D) Water stress in the proposed model

[Figure]

Figure 8. Sensitivity analysis for available water.

*Comment 3-3: **Third, the use of significant numbers, and exponents: in this type of models, parameters***

***may have a maximum of three significant numbers. So declaring values such as 1880616733 doesn't***

*make any sense. Why not report it as 1,880 x10^6? Or even better, as 1.88 x10^9. Tables 2 and 3 are difficult to interpret because of different exponents used; why not adhere to the convention and use consistently 10^3, 10^6 and 10^9 and not anything in between?*

Response: Thank you for your suggestion. These expressions have been corrected and all others checked in the revised manuscript; for example:

**Table 2.** Optimal solutions to imported and exported crops

| | Agricultural sector | | |
|---|---|---|---|
| | Wheat | Maize | Sunflower |
| Imported crops $IM_k$ ($10^9$ kg) | 1.30 | 0.327 | 1.41 |
| Exported crops $EM_k$ ($10^9$ kg) | 0.781 | 0.730 | 0.208 |
| Virtual water $VW_k \times IM_k$ ($10^9$ m$^3$) | 1.21 | 0.108 | 2.84 |
| Virtual land $IM_k/y_k$ (hm$^2$) | 242,456 | 23,630 | 544,853 |

**Table 3.** Optimal water withdrawal results ($10^9$ m$^3$)

| Sectors | Agricultural sector | | | Industrial sector | Domestic sector |
|---|---|---|---|---|---|
| Initial | | $X_1$ | | $X_2$ | $X_3$ |
| water rights | | 3.05 | | 0.015 | 1.38 |
| Water | $X_{11}$ | $X_{12}$ | $X_{13}$ | | |
| irrigation | 0.300 | 0.821 | 0.841 | | |
| Water | | | | WTI | WTD |
| transfer | | -1.09 | | 0.014 | 1.07 |

Comment 3-4: *Tables 4, 5, 7 and 8 could benefit if the first column would simply explain what the parameters (X1 etc.) actually mean, including their units. It is also not clear in these table what the final allocated amount of water is (is the volume of water finally and actually allocated to the agricultural sector X1-WTI-WTD; and for industry X2+WTI and for domestic X3+WTD?). Similar, what is the net import (imports minus exports of a certain crop) value of each of the three crops?*

Response: Thank you for your suggestion. We revised the tables, added several rows to elaborate the final used water from the different sectors and the net import value of each of the three crops, and added simple explanations for the parameters ($X_1$ etc.) in the first column of these tables.

Comment 4-1: *Fourth, not all variables/parameters are properly introduced in the text (and only later the reader is aware of the existence of annexes), and several parameters/variables have wrong or incomplete units/dimensions. For example, what does the variable theta (eq. 1) physically mean? What*

*does it mean if its value increases from 0.5 to say 0.8 and if it decreases to 0.3? Similar for vulnerability F (section 5.4.2). And what does "destroying degree (caused by deficient water withdrawal)" (line 498) mean; and how does it compare/contrast with "economic loss (caused by excess water withdrawal)" (line 499)?*

Response: Thank you for your valuable comments. We have reorganized the paper structure and added a more detailed explanation for the parameters' physical meanings, which are highlighted in the revised manuscript.

In this paper, the $\theta$ refers to the degree of willingness to pay by the two trading participants; therefore, $\theta = 0.5$ represents an average of willingness to pay by the two trading participants. As $\theta$ increases, the price tends towards the buyer's price and as $\theta$ decreases, the price tends to the seller's price. The description for vulnerability was also modified from "destroying degree (caused by deficient water withdrawal)" to "demand loss degree", which refers to an inability to satisfy the water demands, and changed "economic loss (caused by excess water withdrawal)" to "supply loss degree", which refers to losses from excess water supplies.

*Comment 4-2: The correct unit/dimension of $ERP_k$ is not $RMB/m^3$ but I think it should be RMB/kg. The correct unit/dimension of $W_k$ is not $m^3$ but $m^3/hm^2$. All fluxes should have a time dimension, such as $Xi = m^3/yr$; but also evaporation ET, rainfall R, irrigation w; and even crop consumption (kg per person per year!)). What is the difference between $w_k$ (water irrigation for crop k) and $x_{1k}$ (water irrigated to crop k in the agricultural sector)? What are beta ☐and $P_k$ (eq. C5)? Why is effective rainfall crop dependent ($R_k$)? Equation C3 is the well-known Penman equation and need not be reproduced here. A reference would suffice.*

Response: Thank you for your detailed comments. We have corrected the units such as $ERP_{k'}$ $W_k$, and added a time dimension to all fluxes. $w_k$ (water irrigation for crop k) is used to predict the agricultural water demand using Equations (A1)-(A4), and $x_{1k}$ (irrigation water for crop k in the agricultural sector) is the decision variable, which is solved using the proposed model. $\beta$ is the coefficient for efficient rain, and is the proportion of total rain that infiltrates the soil profile and does not contribute to deep percolation, while $P_k$ is the actual rain. The accumulated effective rain for each crop is different because of the different growth

periods for each crop.

 ***Fifth, what do you mean with the following often repeated sentence: "inter-regional exports and international imports" (lines 475, 478, 504, 521); does this mean that only export between regions within China are considered and not internationally, and that only imports from outside China are considered? If so, why? If not, change the formulation.***

Response: Thank you for your detailed comment. International imports are goods from other countries and reflect the global virtual water trade. Inter-regional exports were also considered as one of the means for providing agricultural food to southern China. Because of its comparative land advantage (Zhao et al., 2019), the Hetao irrigation district has a duty to satisfy the southern China demand for crops. Further, as the Hetao irrigation district is a small typical crop growing area, excess crops are generally exported to other provinces in China rather than being exported to other countries. Therefore, the incorporation of the blue and virtual water transfers varies depending on the target of the studies. In the revised version, we focused on a small irrigation district and characterized the virtual water transfers from an inter-regional export perspective due to the small crop yields. If global data were available in the future, such as all crop yields from all irrigation districts in China as well as information on the demand for crops in other countries, a new optimization model could be established that considers the global virtual water transfers from a global perspective, and explores bilateral trade under changing hydrological conditions.

Response: Thank you for your comment. We have revised figure 4.

[Figure]

Figure 4. Steps for solving the proposed model

Comment 10: **Why didn't the authors refer to their own recent publication (Xu, Yao, Zhou, Moudi and Zhang, 2019, in the Journal of Hydrology), since it also uses the Stackelberg approach, but in a different manner? In what ways does it differ?**

Response: Thank you for your suggestion. To solve a water allocation problem that included blue and virtual transfers, this study sought to achieve a Stackelberg-Nash-Harsanyi equilibrium. However, Xu et al. (2019) sought to optimize the water allocation in an irrigation district for water rights transactions based on a Stackelberg-Nash-Cournot equilibrium model. Therefore, the main differences between the two papers are that Xu et al. (2019) did not consider the import or export of crops, which must be included when optimizing a water allocation problem in a water-scarce area as virtual water transfers allows for the redistribution of water resources between countries/regions, which means that water-scarce countries are able to conserve their own water resources, but water-sufficient countries are able to obtain greater economic benefits by selling water-intensive goods. By incorporating the blue and virtual water transfers, the proposed model is able to offers further insights into crop planting and import/export quantities.

**A list of all relevant changes**

1. Introduction and Conclusion sections have been condensed.

2. An academic editor has been asked to review and polish the revised manuscript.

3. The revised manuscript has been reorganized in order to avoid repetition.

4. Figures have been revised to make them more readable.

5. Tables have been revised to unify the use of significant numbers, and exponents.

6. Sensitivity Analysis section have been rewritten to demonstrate the robustness and plausibility of the model.

**A novel data-driven analytical framework on hierarchical water allocation integrated with blue and virtual water transfers**

Liming Yao[1,2], Zhongwen Xu[1], Huijuan Wu[3], and Xudong Chen[1]

[1]Business School, Sichuan University, Chengdu 610064, China
[2]State Key Laboratory of Hydraulics and Mountain River Engineering, Sichuan University, Chengdu, 610064, China
[3]Institute of Water Policy, Lee Kuan Yew School of Public Policy, National University of Singapore, 469A Bukit Timah Road, 259770, Singapore

**Correspondence:** Xudong Chen (chenxudong198401@163.com)

**Abstract.** In this study a novel data-driven analytical framework is proposed for cooperative strategies that ensure the optimal allocation of blue and virtual water transfers under different hydrological and economic conditions. A Stackelberg-Nash-Harsanyi equilibrium model is also developed to deal with the hierarchical conflicts between the water affairs bureau and multiple water usage sectors and overcome problems associated with water scarcity and uneven distribution. It was found that cooperative blue and virtual water transfer strategies could save water and improve utilization efficiency without harming sector benefits or increasing the ecological stress. Data-driven analyses were employed to simulate the hydrological and economic parameters, such as available water, crop import price and water market price under various polices. By adjusting the hydrological and economic parameters, it was found that the optimal allocation and transfer strategies were more sensitive to hydrological factors than economic factors. It was also found that cooperative blue/virtual water transfers respond to market fluctuations. Overall, the proposed framework provides sustainable management for physical and virtual water supply systems under future hydrological and economic uncertainties.

*Copyright statement.* The authors declare that they have no known competing financial interests or personal relationships that could have appeared to influence the work reported in this paper.

[revised manuscript text omitted]
 \\ \min F_1 = \omega^{\text{DD}} F_1^{\text{DD}} + \omega^{\text{EL}} F_1^{\text{EL}} \\ s.t. \begin{cases} F_1^{\text{DD}} = \frac{1}{3} \frac{\sum_{k=1}^{3} \max((d_{1k} - x_{1k}), 0)}{\sum_{k=1}^{3} d_{1k}} \\ F_1^{\text{EL}} = \frac{1}{3} \frac{\sum_{k=1}^{3} \max((x_{1k} - d_{1k}), 0)}{\sum_{k=1}^{3} d_{1k}} \\ \sum_{k=1}^{s} A_k \leq A, A_k = \frac{X_{1k}}{W_k} \\ x_{ik} > 0 \end{cases} \\ \min F_2 = \omega^{\text{DD}} F_2^{\text{DD}} + \omega^{\text{EL}} F_2^{\text{EL}} \\ s.t. \begin{cases} F_2^{\text{DD}} = \frac{\max(d_2 - X_2, 0)}{d_2} \\ F_2^{\text{EL}} = \frac{\max(X_2 - d_2, 0)}{d_2} \\ X_2 > 0 \end{cases} \\ \min F_3 = \omega^{\text{DD}} F_3^{\text{DD}} + \omega^{\text{EL}} F_3^{\text{EL}} \\ s.t. \begin{cases} F_3^{\text{DD}} = \frac{\max(d_3 - X_3, 0)}{d_3} \\ F_3^{\text{EL}} = \frac{\max(X_3 - d_3, 0)}{d_3} \\ X_3 > 0 \end{cases} \end{cases}$$

[revised manuscript text omitted]

---

## Editor Decision (ED1)

A novel data-driven analytical framework on hierarchical water allocation integrated with blue and virtual water transfers

The manuscript has been reviewed by two reviewers, and the authors have, in my view, responded adequately to the issues raised.

I have, however, additional comments that were not raised by the reviewers, and which I like to share with the authors, and I invite them to react to these. My comments are quite significant. When the authors submit a revised manuscript I may decide to send it again out for review.

The paper addresses an important and interesting issue, namely how to deal with water scarcity taking both blue water transfers and virtual water transfers into account, in a 2-level decision setting, and also taking account of three different water using sectors. Quite a complex setting, and an ambitious undertaking. Given such complexity it is important that the argument is clearly presented and here the manuscript needs to be improved significantly (towards the end of this comment I give details what in my view should be addressed to improve the readability of the paper).

A first comment is that the paper makes no attempt whatsoever to validate the proposed model, or at least to show that model outcomes are plausible; this could have been done, for example by comparing model results with observed data, and discussing similarities and differences. Similarly, the authors could have conducted sensitivity analyses to verify how sensitive model results are to changes in input values of certain critical parameters; not for the authors to draw far reaching conclusions, but rather to re-assure the reader of the validity of the model. In the current manuscript the authors have indeed conducted several sensitivity analyses (on water availability, sectoral water demands, prices of import crops and water price), and they draw far reaching conclusions on the outcomes, without even trying to explain these outcomes (see below). But this is not convincing to me. It would be much more convincing to use sensitivity analysis to demonstrate the robustness and plausibility of the model.
The comparison of model results with scenarios omitting virtual water transfers and blue water transfers and with the two-staged model (section 5.2) is interesting, although it is not clear to me how the proposed model differs from the two-stage model. I would suggest to use this comparison to demonstrate the validity (and value) of the proposed model; rather than to formulate far-reaching conclusions. First the model must be validated before it is used to draw conclusions.

A second comment is that the entire section 5.4 on sensitivity analyses raises more questions than it answers. For example, consider the available water (Table 5): if I understand it well, when there is 10% more water available, less water is allocated to agriculture (X1), and when there is 15% more water than the base case, more water is allocated to agriculture, and when there is even more water (20% more than the base case) the volume allocated to agriculture is suddenly halved. This begs for an explanation. Similar questions may be raised concerning the water allocated to the domestic sector (X3). When considering changes in the water price, it is concluded that the price elasticity of water transaction is not linear, as is clearly displayed in Figure 12. But the authors fail to explain why this is so: is this because the model is working as it is supposed to, or may there be something wrong with the model? These are just a few examples on the problematic nature of this section 5.4. As stated in my first comment: I would prefer that the sensitivity analyses are used to validate the model rather than anything else.

This also means that the concluding section can be significantly shortened (as the current conclusions spend quite some words on findings from section 5.4).

The third comment: I find the manuscript not easy to read. In fact it was for me quite cumbersome to read.

First, the English is often grammatically incorrect, unclear or oddly formulated. This needs to be addressed. I advise the authors to engage a native English-speaking academic to carefully check the language used.

Second, some figures were for me impossible to interpret. Consider Figure 5: I guess this is a flow diagram indicating the source and destination of the three crops considered; but I couldn't understand it. Figures 8 and 9 I failed to understand at all. The vertical axis of Figure 11 remains a mystery to me. In all figures, the axis should not only have numbers but also the units declared. E.g. what does the negative values in Figure 10d mean?

Third, the use of significant numbers, and exponents: in this type of models, parameters may have a maximum of three significant numbers. So declaring values such as 1880616733 doesn't make any sense. Why not report it as 1,880 x$10^6$? Or even better, as 1.88 x$10^9$. Tables 2 and 3 are difficult to interpret because of different exponents used; why not adhere to the convention and use consistently $10^3$, $10^6$ and $10^9$ and not anything in between?
Tables 4, 5, 7 and 8 could benefit if the first column would simply explain what the parameters (X1 etc.) actually mean, including their units. It is also not clear in these table what the final allocated amount of water is (is the volume of water finally and actually allocated to the agricultural sector X1-WTI-WTD; and for industry X2+WTI and for domestic X3+WTD?). Similar, what is the net import (imports minus exports of a certain crop) value of each of the three crops?

Fourth, not all variables/parameters are properly introduced in the text (and only later the reader is aware of the existence of annexes), and several parameters/variables have wrong or incomplete units/dimensions. For example, what does the variable theta $\theta$ (eq. 1) physically mean? What does it mean if its value increases from 0.5 to say 0.8 and if it decreases to 0.3? Similar for vulnerability F (section 5.4.2). And what does "destroying degree (caused by deficient water withdrawal)" (line 498) mean; and how does it compare/contrast with "economic loss (caused by excess water withdrawal)" (line 499)?
The correct unit/dimension of ERPk is not RMB/$m^3$ but I think it should be RMB/kg. The correct unit/dimension of Wk is not $m^3$ but $m^3$/$hm^2$. All fluxes should have a time dimension, such as Xi = $m^3$/yr; but also evaporation ET, rainfall R, irrigation w; and even crop consumption (kg per person per year!)). What is the difference between wk (water irrigation for crop k) and x1k (water irrigated to crop k in the agricultural sector)? What are beta $\beta$ and Pk (eq. C5)? Why is effective rainfall crop dependent (Rk)? Equation C3 is the well-known Penman equation and need not be reproduced here. A reference would suffice.

Fifth, what do you mean with the following often repeated sentence: "inter-regional exports and international imports" (lines 475, 478, 504, 521); does this mean that only export between regions within China are considered and not internationally, and that only imports from outside China are considered? If so, why? If not, change the formulation.

Minor comments:

- The paper is very long. Some of the equations and Tables that now appear in the main text can go, together with the Appendices, to Supplementary Materials.

- The introduction is very long, and can in my view be shortened

- The paper states that wheat and sunflower need more water compared to maize. I agree that sunflower generally requires more water than maize and wheat; but generally wheat needs a similar volume of water as maize, or slightly less.

- In Figure 4, I guess that one horizontal arrow pointing from the lower level (left) to the upper level (right) is missing; without it there is no dynamic between both levels.

- Why didn't the authors refer to their own recent publication (Xu, Yao, Zhou, Moudi and Zhang, 2019, in the Journal of Hydrology), since it also uses the Stackelberg approach, but in a different manner? In what ways does it differ?

---

## Author Response (AR2)

Dear editor,

Thank you for your time and the detailed suggestions. We have revised the minor technical issues carefully.

Xudong Chen

**A novel data-driven analytical framework on hierarchical water allocation integrated with blue and virtual water transfers**

Liming Yao[1,2], Zhongwen Xu[1], Huijuan Wu[3], and Xudong Chen[1]

[1]Business School, Sichuan University, Chengdu 610064, China
[2]State Key Laboratory of Hydraulics and Mountain River Engineering, Sichuan University, Chengdu, 610064, China
[3]Institute of Water Policy, Lee Kuan Yew School of Public Policy, National University of Singapore, 469A Bukit Timah Road, 259770, Singapore

**Correspondence:** Xudong Chen (chenxudong198401@163.com)

**Abstract.** In this study a novel data-driven analytical framework is proposed for cooperative strategies that ensure the optimal allocation of blue and virtual water transfers under different hydrological and economic conditions. A Stackelberg-Nash-Harsanyi equilibrium model is also developed to deal with the hierarchical conflicts between the water affairs bureau and multiple water usage sectors and overcome problems associated with water scarcity and uneven distribution. It was found

5  that cooperative blue and virtual water transfer strategies could save water and improve utilization efficiency without harming sector benefits or increasing the ecological stress. Data-driven analyses were employed to simulate the hydrological and economic parameters, such as available water, crop import price and water market price under various policies. By adjusting the hydrological and economic parameters, it was found that the optimal allocation and transfer strategies were more sensitive to hydrological factors than economic factors. It was also found that cooperative blue/virtual water transfers respond to market

10  fluctuations. Overall, the proposed framework provides sustainable management for physical and virtual water supply systems under future hydrological and economic uncertainties.

*Copyright statement.* The authors declare that they have no known competing financial interests or personal relationships that could have appeared to influence the work reported in this paper.

[revised manuscript text omitted]
 \\[2mm] \min F_1 = \omega^{\text{DD}} F_1^{\text{DD}} + \omega^{\text{EL}} F_1^{\text{EL}} \\[2mm] s.t. \begin{cases} F_1^{\text{DD}} = \frac{1}{3} \frac{\sum\limits_{k=1}^{3} \max((d_{1k} - x_{1k}), 0)}{\sum\limits_{k=1}^{3} d_{1k}} \\[4mm] F_1^{\text{EL}} = \frac{1}{3} \frac{\sum\limits_{k=1}^{3} \max((x_{1k} - d_{1k}), 0)}{\sum\limits_{k=1}^{3} d_{1k}} \\[4mm] \sum\limits_{k=1}^{s} A_k \leq A, A_k = \frac{X_{1k}}{W_k} \\[3mm] x_{ik} > 0 \end{cases} \\[2mm] \min F_2 = \omega^{\text{DD}} F_2^{\text{DD}} + \omega^{\text{EL}} F_2^{\text{EL}} \\[2mm] s.t. \begin{cases} F_2^{\text{DD}} = \frac{\max(d_2 - X_2, 0)}{d_2} \\[2mm] F_2^{\text{EL}} = \frac{\max(X_2 - d_2, 0)}{d_2} \\[2mm] X_2 > 0 \end{cases} \\[2mm] \min F_3 = \omega^{\text{DD}} F_3^{\text{DD}} + \omega^{\text{EL}} F_3^{\text{EL}} \\[2mm] s.t. \begin{cases} F_3^{\text{DD}} = \frac{\max(d_3 - X_3, 0)}{d_3} \\[2mm] F_3^{\text{EL}} = \frac{\max(X_3 - d_3, 0)}{d_3} \\[2mm] X_3 > 0 \end{cases} \end{cases} \tag{
[revised manuscript text omitted]

---

## Editor Decision (ED2)

[revised manuscript text omitted]
 \\[2mm] \min F_1 = \omega^{\text{DD}} F_1^{\text{DD}} + \omega^{\text{EL}} F_1^{\text{EL}} \\[2mm] s.t. \begin{cases} F_1^{\text{DD}} = \dfrac{1}{3}\dfrac{\sum_{k=1}^{3}\max((d_{1k}-x_{1k}),0)}{\sum_{k=1}^{3} d_{1k}} \\[4mm] F_1^{\text{EL}} = \dfrac{1}{3}\dfrac{\sum_{k=1}^{3}\max((x_{1k}-d_{1k}),0)}{\sum_{k=1}^{3} d_{1k}} \\[4mm] \displaystyle\sum_{k=1}^{s} A_k \leq A, A_k = \dfrac{X_{1k}}{W_k} \\[3mm] x_{ik} > 0 \end{cases} \\[2mm] \min F_2 = \omega^{\text{DD}} F_2^{\text{DD}} + \omega^{\text{EL}} F_2^{\text{EL}} \\[2mm] s.t. \begin{cases} F_2^{\text{DD}} = \dfrac{\max(d_2-X_2,0)}{d_2} \\[2mm] F_2^{\text{EL}} = \dfrac{\max(X_2-d_2,0)}{d_2} \\[2mm] X_2 > 0 \end{cases} \\[2mm] \min F_3 = \omega^{\text{DD}} F_3^{\text{DD}} + \omega^{\text{EL}} F_3^{\text{EL}} \\[2mm] s.t. \begin{cases} F_3^{\text{DD}} = \dfrac{\max(d_3-X_3,0)}{d_3} \\[2mm] F_3^{\text{EL}} = \dfrac{\max(X_3-d_3,0)}{d_3} \\[2mm] X_3 > 0 \end{cases} \end{cases}$$

[revised manuscript text omitted]